# Concomitant medication, comorbidity and survival in patients with breast cancer

Elise Dumas [1,2,3], Beatriz Grandal Rejo[1], Paul Gougis[1], Sophie Houzard[4], Judith Abécassis [1,5], Floriane Jochum[1,6], Benjamin Marande[1], Annabelle Ballesta[7], Elaine Del Nery [8], Thierry Dubois [9], Samar Alsafadi [10], Bernard Asselain[11], Aurélien Latouche [2,7,12], Marc Espie[13], Enora Laas[14], Florence Coussy[15], Clémentine Bouchez[13], Jean-Yves Pierga[15], Christine Le Bihan-Benjamin[4], Philippe-Jean Bousquet [16,17], Judicaël Hotton[18], Chloé-Agathe Azencott [2,3,19], Fabien Reyal [1,14,18] ✉ & Anne-Sophie Hamy[1,15]

Between 30% and 70% of patients with breast cancer have pre-existing chronic conditions, and more than half are on long-term non-cancer medication at the time of diagnosis. Preliminary epidemiological evidence suggests that some non-cancer medications may affect breast cancer risk, recurrence, and survival. In this nationwide cohort study, we assessed the association between medication use at breast cancer diagnosis and survival. We included 235,368 French women with newly diagnosed non-metastatic breast cancer. In analyzes of 288 medications, we identified eight medications positively associated with either overall survival or disease-free survival: rabeprazole, alverine, atenolol, simvastatin, rosuvastatin, estriol (vaginal or transmucosal), nomegestrol, and hypromellose; and eight medications negatively associated with overall survival or disease-free survival: ferrous fumarate, prednisolone, carbimazole, pristinamycin, oxazepam, alprazolam, hydroxyzine, and mianserin. Full results are available online from an interactive platform (https://adrenaline.curie.fr). This resource provides hypotheses for drugs that may naturally influence breast cancer evolution.

Breast cancer (BC) is the most frequent cancer in women and the leading cause of cancer deaths in women worldwide. Its incidence increases with age, as does the incidence of many other chronic diseases, such as hypertension, diabetes, and dyslipidemia. Between 30% and 70% of patients with BC suffer from pre-existing comorbid conditions at BC diagnosis[1–3] and more than half are already taking medication (chronic treatment with non-cancer drugs)[4,5].

There is a strong complex interplay between comorbid conditions and concomitant medication, but most previous studies on this topic have focused on either comorbid conditions[1,3,6,7] or concomitant medication[4,8]. Several studies have reported epidemiological evidence for associations between non-cancer treatments, such as aspirin or non-steroidal anti-inflammatory drugs, and lower BC risk[9]. Other

medications, such as statins[10,11], beta-blockers[12,13], and metformin[11], have been shown to be associated with lower rates of BC recurrence or better survival after BC. Conversely, several frequently prescribed medications have been shown to increase the risk of BC[14,15] or to interact with BC treatments[16–19].

In France, all the medical and administrative information relating to the reimbursement of healthcare expenses is collected and aggregated within the National Health Data System[20] (*Système National des Données de Santé*, SNDS). The SNDS is among the largest and most exhaustive health data resources worldwide, covering approximately 70 million people in considerable detail. It provides a wealth of information that can be used to generate real-world evidence to develop precision medicine or support public health decision-making[21,22].

We hypothesize that several concomitant medications taken prior to diagnosis may modify the natural course of BC. The main objective of this study designated ADRENALINE (Atlas of Drugs, Comorbidities, and Cancer Treatment Survival Interaction), is to analyze the impact of medication use in the six months preceding the diagnosis of BC on overall survival (OS, main outcome), and disease-free survival (DFS, secondary outcome) in a very large cohort of French women diagnosed with BC. Using SNDS data for a published cohort of patients with BC[23], we identified all the non-cancer medications commonly prescribed in the six months preceding BC diagnosis. We adjusted for approximately a hundred confounding factors to identify medications with a significant positive or negative association with DFS or OS. We then used mediation analyzes to estimate the extent to which these associations could be explained by a change in BC subtype or nodal status at diagnosis due to the use of the medications concerned. Finally, we built an interactive tool to explore the distribution of comorbid conditions, medications used at BC diagnosis, their co-occurrence, and the association between these medications and survival (available as a public resource from https://adrenaline.curie.fr).

## Results

### Characteristics of the patients and tumors
The analyzes included 235,368 patients with BC in total, of whom 12.1% relapsed or died and 6.6% died during follow-up. The median follow-up time was 54.6 months for OS and 53.9 months for DFS. Median age at diagnosis was 60 years (Table 1). The distribution of BC subtypes was as follows: luminal (65.1%), TNBC (7.7%), HER2-positive (8.4%) (18.9% undefined tumors). Most patients had node-negative disease (81.2%), received radiotherapy (85.3%) and endocrine therapy (70.4%), and approximately one-third received chemotherapy (38.3%).

### Comorbid conditions
At least one comorbid condition was present at diagnosis in 47.0% of patients. The frequency and number of comorbid conditions increased with age and deprivation index (Supplementary Fig. 1). Cardiovascular diseases were the most frequent (25.6% of patients), followed by endocrine/metabolic diseases (21.9%) and psychiatric (12.9%) disorders (Table 1). The top three comorbid diagnoses were hypertension (20.5%), diabetes (8.3%) and obesity (8.2%) (Fig. 1). Strong associations were found between certain comorbid conditions, such as hypertension and diabetes ($n = 10,760$), or hypertension and dyslipidemia ($n = 10,643$, Supplementary Fig. 2). An interactive viewer of associations between comorbid conditions is available online (Supplementary Fig. 3, https://adrenaline.curie.fr/static/network_comor/index.html). No comorbid condition was found in 53.0% of patients, some of whom were taking non-specific medications, such as vitamins (21.1%), analgesics (14.1%), or sex hormones and modulators of the genital system (15.4%) (Supplementary Fig. 4).

### Concomitant medication
Approximately three quarters of patients were on at least one concomitant medication in the six months preceding diagnosis (76.0%), and the frequency and number of medications increased with age and deprivation index (Supplementary Fig. 5). The three main anatomical classes were drugs targeting the alimentary tract and metabolism (ATC A), the cardiovascular system (ATC C) and the nervous system (ATC N), with colecalciferol, paracetamol, and levothyroxine the three medications most frequently reported (Fig. 2, Supplementary Data 1, adrenaline.curie.fr/comed_description). Some medications were often prescribed together (Supplementary Fig. 6): vitamins (A11) and mineral supplements (A12, $n = 13,918$); diuretics (C03) and agents acting on the renin-angiotensin system (C09, $n = 26,456$); or psycholeptics (N05) and psychoanaleptics (N06, $n = 16,706$). Co-prescriptions can be explored further with an online interactive tool (Supplementary Fig. 7, https://adrenaline.curie.fr/static/network_comor/index.html).

The number of concomitant medications taken at diagnosis was correlated with the number of comorbid conditions (Supplementary Fig. 8), and there was a strong association between comorbid conditions and concomitant medications from the corresponding therapeutic class (Supplementary Fig. 4), such as cardiovascular conditions and agents acting on the renin-angiotensin system (55.0%).

### Association between concomitant medication and survival
The 288 medications selected for the analyzes included 113 (39%) that passed the adjustment quality test (Figs. 3 and 4, Supplementary Data 2, 3). Before adjustment for multiple tests, 32 medications were significantly associated with OS ($n = 25$, Fig. 5A), or DFS ($n = 23$, Fig. 6A), including 16 medications associated with both OS and DFS: rabeprazole, alverine, ferrous fumarate, atenolol, simvastatin, rosuvastatin, prednisolone, oxazepam, bromazepam, hydroxyzine, mianserin, duloxetine, estriol (vaginal or transmucosal), nomegestrol, pristinamycin, and hypromellose. Among these 16 drugs, all medications targeting the alimentary tract and metabolism (rabeprazole, alverine) were protective, as were medications targeting the cardiovascular system (atenolol, simvastatin, and rosuvastatin) and the genitourinary system (estriol vaginal or transmucosal, nomegestrol). All medications targeting the blood and blood-forming organs (ferrous fumarate) and hormonal system (prednisolone) were deleterious, as were antibiotics (pristinamycin) (Figs. 5A, 6A). For medications targeting the nervous system, we observed mixed-class effects in the association between survival and the intake of benzodiazepines, with bromazepam being protective (HR OS: 0.89, 95% CI: 0.80 to 0.98; HR DFS: 0.91 95% CI: 0.84 to 0.99) whereas oxazepam (HR OS: 1.27, 95% CI: 1.12 to 1.44; HR DFS: 1.20, 95% CI: 1.07 to 1.34) was deleterious.

### Mediation analyzes
The 32 medications significantly associated with a decrease or increase in DFS or OS were selected for mediation analyzes, in which the observed associations were broken down into: (1) direct effect, including the inherent effect of the medication and any effect through other pathways not involving a difference in BC subtype or nodal status, (2) indirect effect through pathways involving a difference in BC subtype; (3) indirect effect through pathways involving a difference in nodal status. The associations between concomitant medication and survival were generally almost entirely attributable to direct effects (Figs. 5B and 6B). Among the 16 medications significantly associated with both OS and DFS, a significant percentage of the estimated ATE could be attributed to differences in tumor subtype for rabeprazole (7.5% non-significant for OS; 47.6% for DFS), rosuvastatin (12.4% for OS; 11.7% for DFS), bromazepam (33.1% for OS; 64.5% for DFS), and hypromellose (22.6% non-significant for OS; 33.6% for DFS); and a significant percentage of the estimated ATE could be attributed to differences in nodal status at diagnosis for alverine (4.4% non-significant for OS; 8.9% for DFS), simvastatin (3.1% for OS; 3.6% for DFS), rosuvastatin (2.3% for OS; 3.1% for DFS), oxazepam (4.4% non-significant for OS; 7.1% for DFS), hydroxyzine (8.6% for OS; 6.8% for DFS), estriol vaginal or transmucosal (7.5% non-significant for OS; 23.8% for DFS), and hypromellose (12.8% for OS; 6% non-significant for DFS). For atenolol, we observed a protective association with survival overall (HR OS 0.77 95% CI 0.65 to 0.90; HR DFS 0.82 95% CI 0.70 to 0.97), but atenolol use was associated with more frequent lymph node involvement at BC diagnosis (−5.4% of the effect).

### Impact on survival after adjustment for multiple testing
After adjustment for multiple testing, sixteen medications remained significantly associated with an increase ($n = 8$) or decrease ($n = 8$) in OS (Fig. 7) or DFS (Fig. 8), six of which were associated with both OS and DFS (simvastatin, rosuvastatin, nomegestrol, prednisolone, pristinamycin, and oxazepam). Rabeprazole (HR OS: 0.77, 95% CI: 0.65 to 0.91), alverine (HR OS: 0.78, 95% CI: 0.67 to 0.91), atenolol

**Table 1 | Characteristics of the patients in the total population, patients without medication at the time of BC diagnosis, and patients on at least one medication at the time of BC diagnosis**

| Category | Variable<br><br>n (%) | Class | Total<br><br>235 368 (100%) | No concomitant medication<br>56 510 (24%) | At least one drug<br>178 858 (76%) |
|---|---|---|---|---|---|
| **Pre-exposure covariates** | | | | | |
| Socio-demographic | Age at diagnosis (years) | | 60.0 [50.0, 69.0] | 52.0 [45.0, 61.0] | 63.0 [53.0, 71.0] |
| | Age at diagnosis (years, classes) | <30 | 1 124 (0.5) | 477 (0.8) | 647 (0.4) |
| | | 30-39 | 10 539 (4.5) | 4 631 (8.2) | 5 908 (3.3) |
| | | 40-49 | 43 206 (18.4) | 17 582 (31.1) | 25 624 (14.3) |
| | | 50-59 | 58 003 (24.6) | 18 297 (32.4) | 39 706 (22.2) |
| | | 60-69 | 64 042 (27.2) | 10 994 (19.5) | 53 048 (29.7) |
| | | 70-79 | 39 163 (16.6) | 3 458 (6.1) | 35 705 (20.0) |
| | | 80+ | 19 291 (8.2) | 1 071 (1.9) | 18 220 (10.2) |
| | Deprivation index (quintiles) | 1st quintile (least deprived) | 46 323 (19.7) | 12 369 (21.9) | 33 954 (19.0) |
| | | 2nd quintile | 46 688 (19.8) | 11 657 (20.6) | 35 031 (19.6) |
| | | 3rd quintile | 45 984 (19.5) | 11 021 (19.5) | 34 963 (19.5) |
| | | 4th quintile | 46 183 (19.6) | 10 504 (18.6) | 35 679 (19.9) |
| | | 5th quintile (most deprived) | 45 992 (19.5) | 9 538 (16.9) | 36 454 (20.4) |
| | | Overseas *départements* | 4 198 (1.8) | 1 421 (2.5) | 2 777 (1.6) |
| | GP consultations* | | 5.0 [3.0, 9.0] | 2.0 [1.0, 4.0] | 6.0 [4.0, 10.0] |
| | GP consultations* (classes) | 0 | 12 964 (5.5) | 8 421 (14.9) | 4 543 (2.5) |
| | | 1–5 | 109 539 (46.5) | 38 489 (68.1) | 71 050 (39.7) |
| | | 6–11 | 32 701 (13.9) | 1 308 (2.3) | 31 393 (17.6) |
| | | 12+ | 80 164 (34.1) | 8 292 (14.7) | 71 872 (40.2) |
| | Gynecologist visits** | | 0.0 [0.0, 1.0] | 1.0 [0.0, 1.0] | 0.0 [0.0, 1.0] |
| | Gynecologist visits** (classes) | 0 | 127 785 (54.3) | 28 225 (49.9) | 99 560 (55.7) |
| | | 1 | 65 868 (28.0) | 17 694 (31.3) | 48 174 (26.9) |
| | | 2–3 | 35 290 (15.0) | 8 941 (15.8) | 26 349 (14.7) |
| | | 4+ | 6 425 (2.7) | 1 650 (2.9) | 4 775 (2.7) |
| | Mammographic screening before diagnosis | No | 146 945 (62.4) | 38 843 (68.7) | 108 102 (60.4) |
| | | Yes | 88 423 (37.6) | 17 667 (31.3) | 70 756 (39.6) |
| Comorbid conditions | Comorbid conditions (binary) | No | 124 652 (53.0) | 44 872 (79.4) | 79 780 (44.6) |
| | | Yes | 110 716 (47.0) | 11 638 (20.6) | 99 078 (55.4) |
| | Comorbid condition category | *Cardiovascular* | 60 146 (25.6) | 2 931 (5.2) | 57 215 (32.0) |
| | | *Endocrine and metabolism* | 51 588 (21.9) | 3 522 (6.2) | 48 066 (26.9) |
| | | *Psychiatric disorders* | 30 372 (12.9) | 4 713 (8.3) | 25 659 (14.3) |
| | | *Frailty (proxy)* | 11 888 (5.1) | 1 181 (2.1) | 10 707 (6.0) |
| | | *Pulmonary* | 10 883 (4.6) | 750 (1.3) | 10 133 (5.7) |
| | | *Rheumatologic disease and connective tissue diseases* | 7 918 (3.4) | 413 (0.7) | 7 505 (4.2) |
| | | *Gastrointestinal* | 7 519 (3.2) | 752 (1.3) | 6 767 (3.8) |
| | | *Neurologic* | 6 983 (3.0) | 746 (1.3) | 6 237 (3.5) |
| | | *Liver* | 2 668 (1.1) | 324 (0.6) | 2 344 (1.3) |
| | | *Kidney* | 2 524 (1.1) | 93 (0.2) | 2 431 (1.4) |
| | | *Other* | 1 015 (0.4) | 103 (0.2) | 912 (0.5) |
| | | *Immune* | 635 (0.3) | 84 (0.1) | 551 (0.3) |
| **Post-exposure covariates** | | | | | |
| BC biology | Inferred BC subtype | luminal | 153 109 (65.1) | 34 117 (60.4) | 118 992 (66.5) |
| | | TNBC | 18 149 (7.7) | 5 532 (9.8) | 12 617 (7.1) |
| | | *HER2+* | 19 722 (8.4) | 5 974 (10.6) | 13 748 (7.7) |
| | | Undefined | 44 388 (18.9) | 10 887 (19.3) | 33 501 (18.7) |
| | Nodal status | Node-negative | 191 164 (81.2) | 45 282 (80.1) | 145 882 (81.6) |
| | | Node-positive | 44 204 (18.8) | 11 228 (19.9) | 32 976 (18.4) |

**Table 1 (continued) | Characteristics of the patients in the total population, patients without medication at the time of BC diagnosis, and patients on at least one medication at the time of BC diagnosis**

| Category | Variable | Class | Total | No concomitant medication | At least one drug |
|---|---|---|---|---|---|
| | n (%) | | 235 368 (100%) | 56 510 (24%) | 178 858 (76%) |
| BC treatment | Breast surgery | Partial mastectomy | 173 173 (73.6) | 40 238 (71.2) | 132 935 (74.3) |
| | | Mastectomy | 62 195 (26.4) | 16 272 (28.8) | 45 923 (25.7) |
| | Radiotherapy | No | 34 683 (14.7) | 8 294 (14.7) | 26 389 (14.8) |
| | | Yes | 200 685 (85.3) | 48 216 (85.3) | 152 469 (85.2) |
| | Chemotherapy | No | 145 116 (61.7) | 29 421 (52.1) | 115 695 (64.7) |
| | | Yes | 90 252 (38.3) | 27 089 (47.9) | 63 163 (35.3) |
| | Endocrine therapy | No | 69 713 (29.6) | 18 628 (33.0) | 51 085 (28.6) |
| | | Yes | 165 655 (70.4) | 37 882 (67.0) | 127 773 (71.4) |

The number of patients, and the percentage of patients (in parentheses), are reported for categorical variables. The median value, and the interquartile range (in parentheses), are reported for continuous variables. *Number of general practitioner (GP) visits in the year preceding BC diagnosis.
**Number of gynecologist visits in the year preceding BC diagnosis.
*GP* general practitioner, *BC* breast cancer, *TNBC* triple-negative breast cancer.

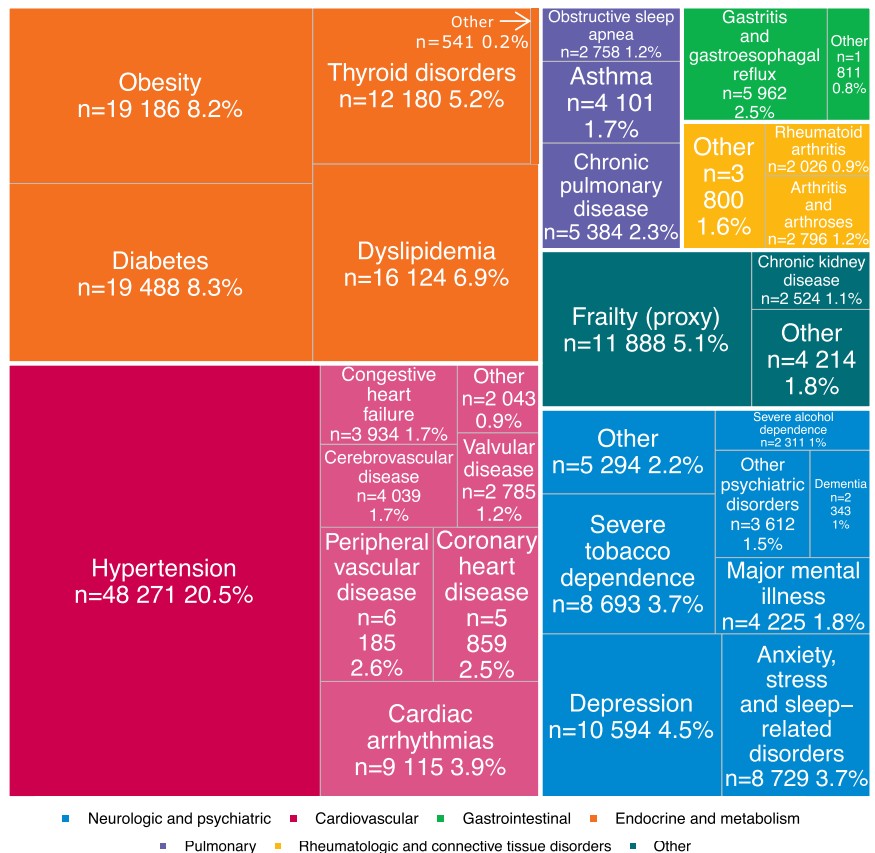

**Fig. 1 | Distribution of comorbid conditions (by disease) in the total population.** Diseases are color-coded by category. Percentages of the total population are reported. In each category, comorbid conditions with fewer than 2000 cases were regrouped into the "Other" category to improve readability. In the neurologic and psychiatric diseases category, "Other" includes anorexia or bulimia (n = 114, 0%), cognitive disabilities (n = 791, 0.3%), epilepsy (n = 1278, 0.5%), hemiplegia, paraplegia or palsy (n = 1600, 0.7%), multiple sclerosis (n = 719, 0.3%), other substance use disorder (n = 244, 0.1%), and Parkinson's disease (n = 989, 0.4%). In the cardiovascular diseases category, "Other" includes coagulopathy (n = 743, 0.3%"), hemoglobinopathy (n = 104, 0%), and pulmonary embolism (n = 1217, 0.5%). In the gastrointestinal diseases category, "Other" includes inflammatory bowel disease (n = 1022, 0.4%), pancreatic disease (n = 232, 0.1%), and peptic ulcer disease (n = 576, 0.2%). In the endocrine and metabolic diseases category, "Other" includes other endocrine disorders (n = 541, 0.2%). In the rheumatologic and connective tissue disorders category, "Other" includes connective tissue diseases (n = 1102, 0.5%), fibromyalgia (n = 324 0.1%), osteoporosis (n = 1817, 0.8%), and rheumatic diseases (n = 664, 0.3%). In the other diseases category, "Other" includes hereditary metabolic disorders (n = 459, 0.2%), myopathies, or disorders of muscles (n = 562, 0.2%), HIV/AIDS (n = 316, 0.1%), organ or tissue transplant (n = 186 0.1%), other immune deficiency (n = 141, 0.1%), chronic hepatitis (n = 1001, 0.4%), cirrhosis (n = 879, 0.4%), and steatosis and hereditary diseases (n = 1046, 0.4%). The data can be further explored on the interactive ADRENALINE web application (https://adrenaline.curie.fr/comorbidity_description). Source data are provided as a Source Data file. Abbreviations: HIV: human immunodeficiency virus; AIDS: acquired immunodeficiency syndrome.

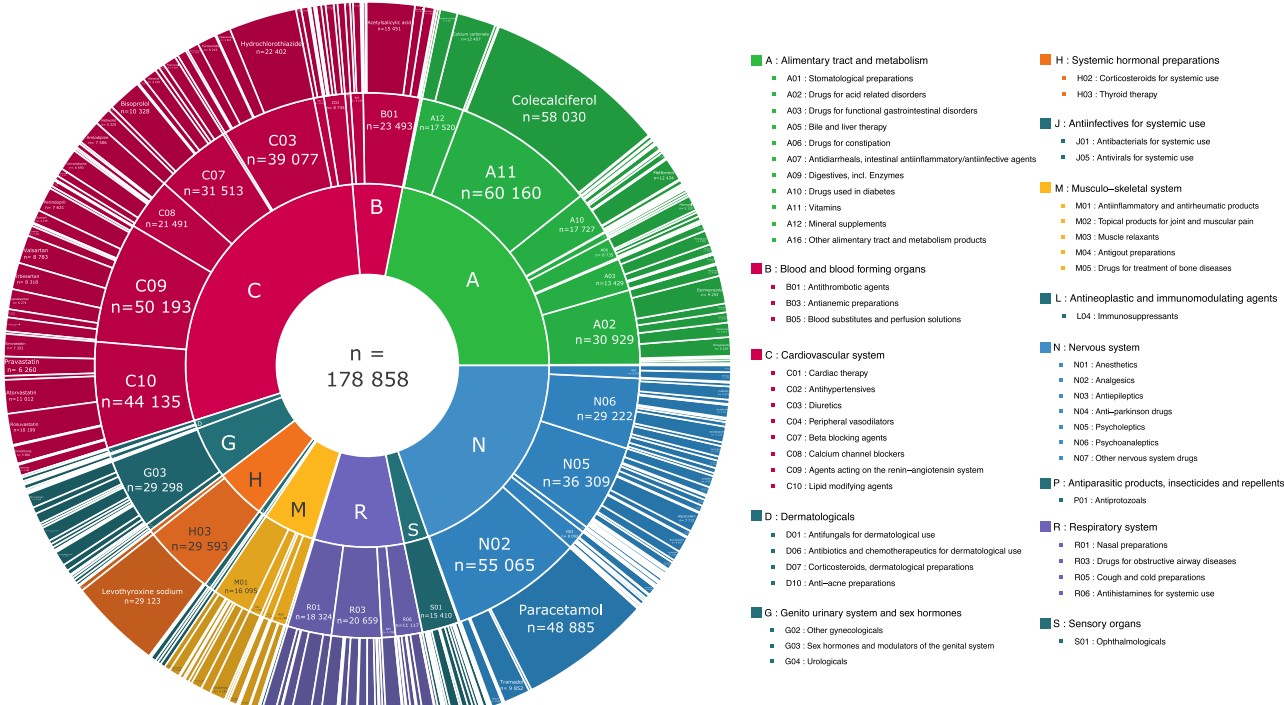

**Fig. 2 | Distribution of concomitant medications by ATC code, for ATC level 1 (inner ring), ATC level 2 (middle ring), and ATC level 5 (outer ring).** Concomitant medications are color-coded by ATC level. Raw data for ATC classes for which the data cannot be read on the graph can be accessed in Supplementary Data 1 or via the interactive display available online at https://adrenaline.curie.fr/comed_description. Source data are provided as a Source Data file. ATC Anatomical Therapeutic Chemical.

(HR OS: 0.77, 95% CI: 0.65 to 0.90), simvastatin (HR OS: 0.73, 95% CI: 0.61 to 0.88; HR DFS: 0.76, 95% CI: 0.63 to 0.93), rosuvastatin (HR OS: 0.64, 95% CI: 0.55 to 0.76; HR DFS: 0.72, 95% CI: 0.63 to 0.82), estriol (vaginal or transmucosal, HR OS: 0.58, 95% CI: 0.42 to 0.80), nomegestrol (HR OS: 0.39, 95% CI: 0.26 to 0.60; HR DFS: 0.74, 95% CI: 0.60 to 0.91), and hypromellose (HR DFS: 0.77, 95% CI: 0.66 to 0.91) were associated with longer survival, whereas ferrous fumarate (HR OS 1.74, 95% CI: 1.26 to 2.40), prednisolone (HR OS: 1.78, 95% CI: 1.18 to 2.69; HR DFS: 1.58, 95% CI: 1.15 to 2.18), carbimazole (HR DFS: 1.42, 95% CI: 1.11 to 1.82), pristinamycin (HR OS: 1.88, 95% CI: 1.37 to 2.58; HR DFS: 1.64, 95% CI: 1.24 to 2.16), oxazepam (HR OS: 1.27, 95% CI: 1.12 to 1.44; HR DFS: 1.20, 95% CI: 1.07 to 1.34), alprazolam (HR DFS: 1.12, 95% CI: 1.04 to 1.20), hydroxyzine (HR DFS: 1.16, 95% CI: 1.05 to 1.29), and mianserin (HR OS: 1.36, 95% CI: 1.14 to 1.61) were associated with shorter survival. We obtained similar results in sensitivity analyzes performed with two different timeframes for the identification of comorbid conditions (Supplementary Table 1).

## Discussion

We performed a large, comprehensive overview of medication taken at the time of BC diagnosis and comorbid conditions, and performed extensive causal inference analyzes to investigate the association between concomitant medication and survival, adjusting for underlying disease and confounding factors. This study provides several new insights.

Little is known about the patterns of comorbid conditions at BC diagnosis, and no gold standard approach has been validated for assessing comorbidity in the cancer context[24]. We analyzed comorbid conditions individually, using a large list of diseases selected from an extensive literature review[25–34]. Consistent with previous studies[35], we found that hypertension, obesity, diabetes, and dyslipidemia were the most frequent conditions, and were strongly associated. Comorbid conditions were strongly linked to concomitant medication intake, but

the overlap was incomplete, probably due to the use of medications unrelated to comorbid conditions (contraceptive pills), or for undefined conditions (analgesics). We developed a causal inference pipeline to isolate the role of concomitant medication from those of the underlying condition and other confounding factors, and to decipher its direct effect on mortality through indirect effects involving modifications in BC subtype or nodal status at diagnosis. We discovered eight protective and eight deleterious associations.

Rabeprazole, a commonly used proton pump inhibitor (PPI), was associated with a 23% decrease in instantaneous risk of death. PPIs are thought to enhance the antitumor effects of chemotherapy[36] and are associated with higher levels of immune infiltration[37]. There is a strong rationale for PPI use as adjuvant anticancer agents, possibly for deacidification of the tumor microenvironment[38–40].

Atenolol, a beta-blocker, was associated with longer overall survival, consistent with the large body of preclinical[41] and clinical[42] data and the findings of meta-analyzes[43]. Multiple relevant mechanisms have been proposed for this effect[44].

Two statins, simvastatin and rosuvastatin, were found to be highly protective, decreasing instantaneous risk of death by 27% and 36% and relapse or death by 24% and 28%, respectively. Statins inhibit the rate-limiting step of cholesterol biosynthesis and lower serum cholesterol concentration, but they also have pleiotropic effects on cell growth, signal transduction, differentiation, and apoptosis, thereby modulating physiological processes essential to cancer initiation and promotion[45]. Statins were found to be associated with a lower risk of BC recurrence[46] and BC-related deaths[47] in observational studies, and are currently under evaluation in two randomized clinical trials (NCT03971019, NCT04601116).

Several sex hormones used either locally (vaginal or transmucosal estriol treatment) or systemically (nomegestrol) were associated with longer survival in our cohort of BC patients when used prior to diagnosis, with HRs of 0.58 and 0.39 for OS, and 0.80 and 0.74 for DFS,

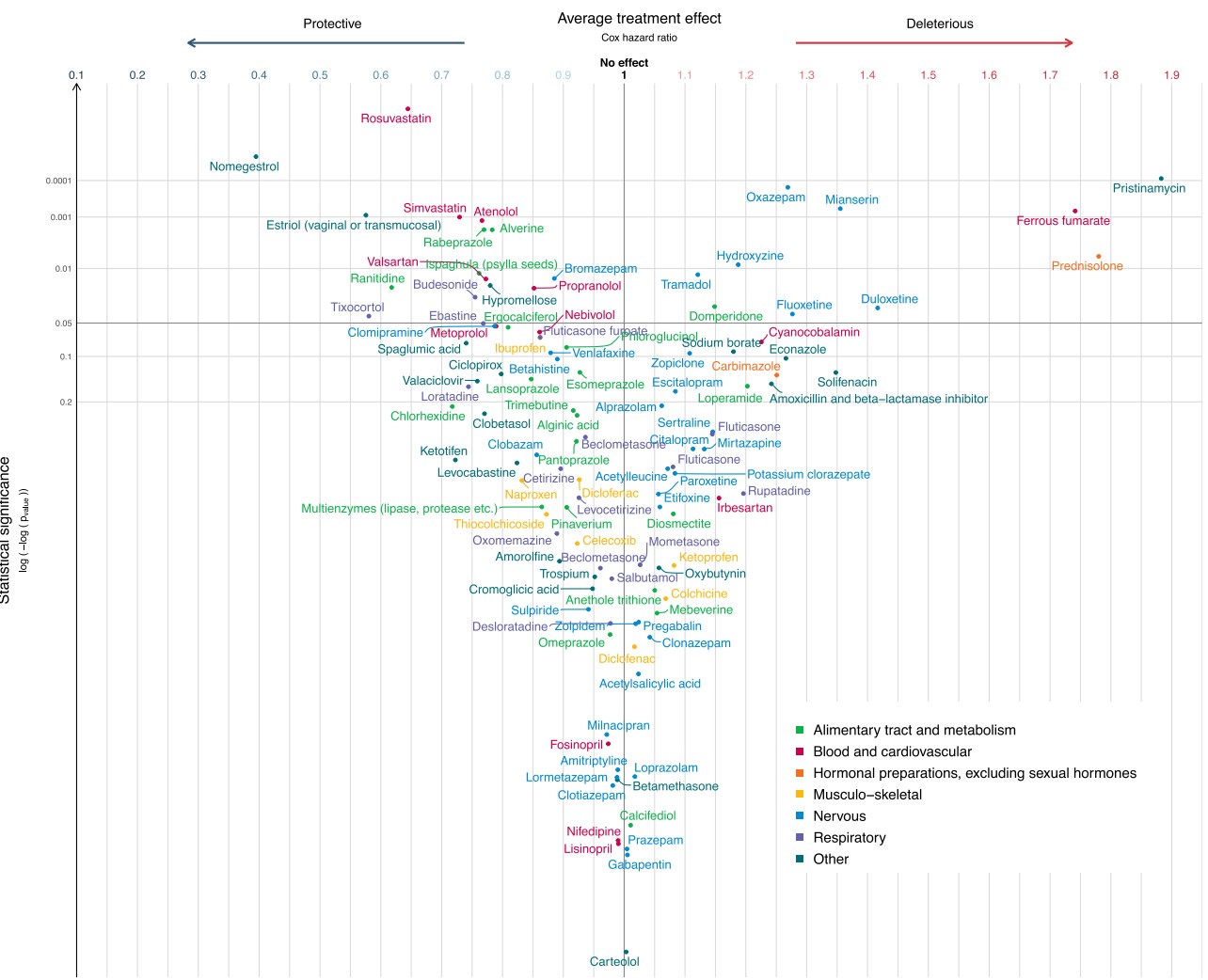

**Fig. 3 | Estimated average treatment effect (ATE) for overall survival (OS) for the 113 medications passing the adjustment quality test.** Medications are represented by circles color-coded by ATC level and linked to the full name of the medication. The ATE (i.e. the Cox hazard ratio, HR) is plotted on the *x*-axis. Lower HRs (protective effect of the medication, increasing overall survival in breast cancer) are displayed on the left. Higher HRs (deleterious effect of the medication, decreasing overall survival in breast cancer) are displayed on the right. An HR of 1 (no effect of the medication on overall survival in breast cancer) is indicated by a vertical line. We used two-sided Wald tests with robust covariances for statistical inference. No adjustment for multiple comparisons was made at this stage of the pipeline. Statistical significance is plotted on the *y*-axis. Lower *p*-values (high statistical significance) are displayed at the top. Higher *p*-values (low statistical significance) are displayed at the bottom. An interactive display is available via the ADRENALINE web application (https://adrenaline.curie.fr/survival_analysis). Source data are provided as a Source Data file. *Estriol (vaginal or transmucosal). ATE average treatment effect, ATC Anatomical Therapeutic Chemical.

respectively. The relationship between sex hormones and BC progression remains incompletely understood. Sex hormones used systematically have previously been associated with an increased risk of BC[48–50], or with an increased risk of relapse when use post-diagnosis[51]. Conversely, post-diagnosis vaginal estrogen therapy has been tentatively associated with a lower risk of BC recurrence[52], specific mortality[53], or all-cause mortality[54], although safety concerns have been raised in patients currently treated with aromatase inhibitors[54]. While our findings do not address the risk of BC incidence or the impact of post-diagnosis hormone use, they suggest that pre-diagnosis sex hormones may be associated with decreased mortality and relapse in BC patients, possibly through changes in tumor biology at diagnosis, as 23.8% of the protective association we observed between vaginal or transmucosal estriol and DFS was mediated by a decreased likelihood of lymph node involvement.

The two remaining protective associations suggested by our results were novel. The use of hypromellose eye drops in the six months prior to BC was associated with a decrease in instantaneous risk of 22% for OS and 23% for DFS. Notably, a significant part of the protective association was mediated by a change in BC subtype and nodal status at BC diagnosis. Alverine use prior to BC diagnosis was also associated with an improved prognosis in our results (HR OS: 0.78; HR DFS: 0.86). Further investigation of these novel observations is warranted.

Eight medications were negatively associated with survival. Preclinical study suggested a role for ferroportin and iron regulation in BC progression and prognosis[55,56]. Here, we found that ferrous fumarate was associated with poorer survival (HR OS: 1.74; HR DFS: 1.39). However, we cannot exclude the possibility of a residual confounding bias due to an underreporting of anemia and low ferritinemia in reimbursement claims. Similarly, we found that two antidepressants, hydroxyzine and mianserin, were associated with decreased survival, but we cannot exclude that these associations could be explained by residual confounding bias due to depression, which has been shown to increase both all-cause and specific mortality in patients with BC[57]. The 42% decrease in instantaneous risk for DFS suggested for carbimazole could also be explained by insufficient adjustment for hyperthyroidism, a condition that may be

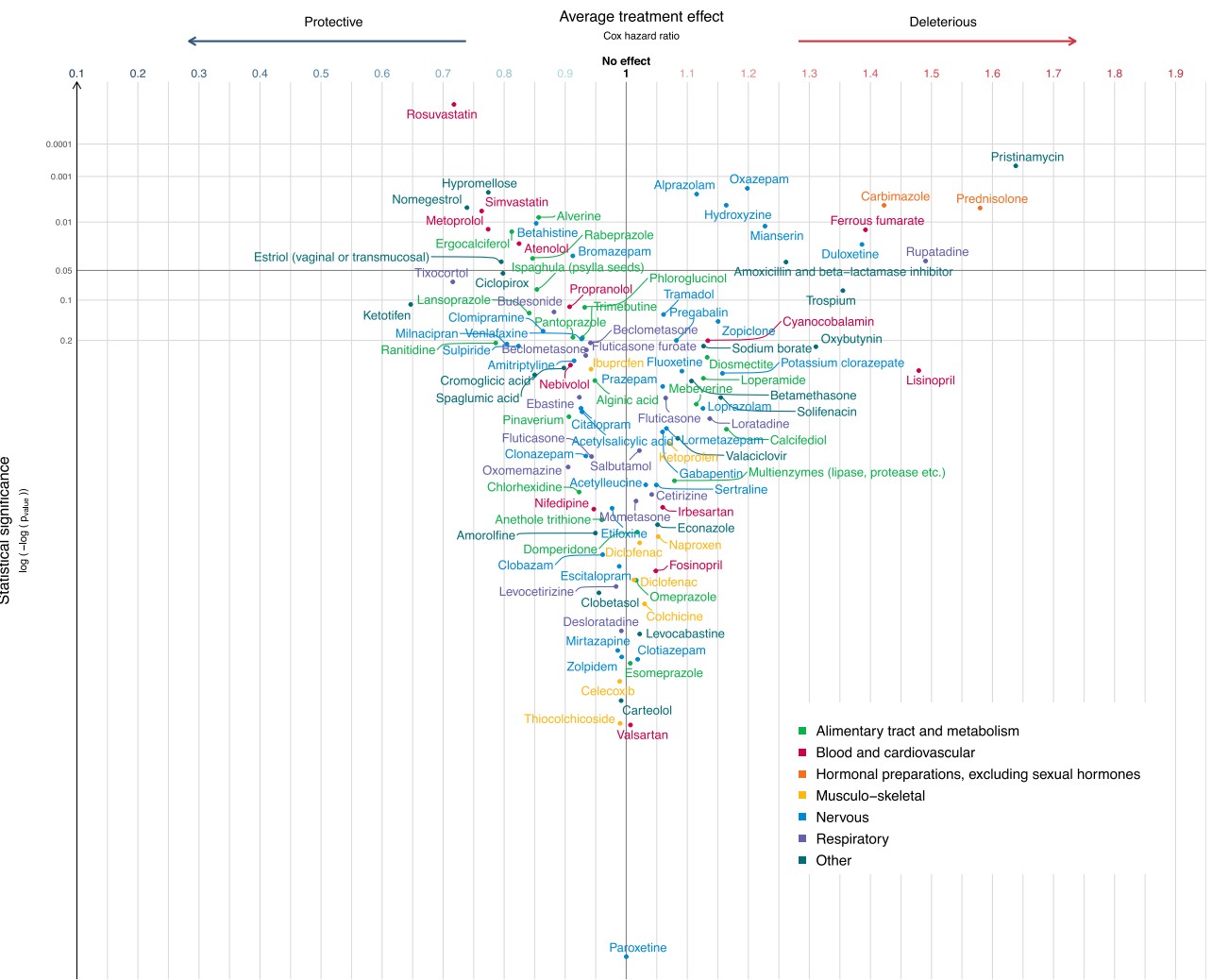

**Fig. 4 | Estimated average treatment effect (ATE) for disease-free survival (DFS) for the 113 medications passing the adjustment quality test.** Medications are represented by circles color-coded by ATC level and linked to the full name of the medication. The ATE (*i.e.* the Cox hazard ratio, HR) is plotted on the x-axis. Lower HRs (protective effect of the medication, increasing disease-free survival in breast cancer) are displayed on the left. Higher HRs (deleterious effect of the medication, decreasing disease-free survival in breast cancer) are displayed on the right. An HR of 1 (no effect of the medication on disease-free survival in breast cancer) is indicated by a vertical line. We used two-sided Wald tests with robust covariances for statistical inference. No adjustment for multiple comparisons was made at this stage of the pipeline. Statistical significance is plotted on the y-axis. Lower p-values (high statistical significance) are displayed at the top. Higher p-values (low statistical significance) are displayed at the bottom. An interactive display is available via the ADRENALINE web application (https://adrenaline.curie.fr/survival_analysis). Source data are provided as a Source Data file. Abbreviations: ATE: average treatment effect; ATC: Anatomical Therapeutic Chemical.

associated with poorer prognosis, possibly through higher baseline mammographic density[58,59].

Pristinamycin, an antibiotic used primarily against staphylococcal and streptococcal infections, was also associated with higher mortality. The negative impact of antibiotics on oncological outcomes – particularly in patients treated with immunotherapy – is attracting growing interest[60], but there is currently no observational data for BC. Our study revealed that prednisolone was associated with a 78% decrease in instantaneous risk for OS and a 58% for DFS, in line with several preclinical studies suggesting that glucocorticoids may promote BC progression and metastasis[61–63], and with an epidemiologic study reporting that the use of glucocorticoids was associated with a decreased risk of stage I-II BCs but an increased risk of stage III-IV BCs[64]. Given the intensive use of glucocorticoids as adjunctive therapy during chemotherapy, we believe that further research on this topic is urgently needed.

Finally, oxazepam and alprazolam were found to be deleterious, whereas other benzodiazepines had no effect (prazepam HR OS: 1.00, 95% CI: 0.86 to 1.17) or a protective effect (bromazepam, HR OS: 0.89,

95% CI: 0.80 to 0.98), suggesting that medications from the same therapeutic class may differentially modify the course of BC. Consistent with our findings, differential effects of benzodiazepines on cancer outcomes have been observed in observational studies of BC risk[65] or pancreatic cancer survival[66], possibly reflecting differences in pharmacological properties and potential interactions with cancer-related pathways[66–68].

We provide here a unique resource, uniting on the same interactive platform an extensive overview of the causal impact of non-oncological medications on a very large, exhaustive cohort of patients with BC. We applied a stringent methodology to minimize confounding bias, and identified sixteen medications affecting relapse or mortality. Our study has several limitations. First, given the very conservative strategy used to correct for multiple testing, we cannot exclude the possibility that several other medications tested had a genuine effect on survival. Second, while most chronic medications are prescription-only drugs, the observed associations for certain medications (*e.g.* paracetamol or vitamins) may be subject to potential mismeasurement bias due to the lack of over-the-counter purchase data in the SNDS. Third, while the

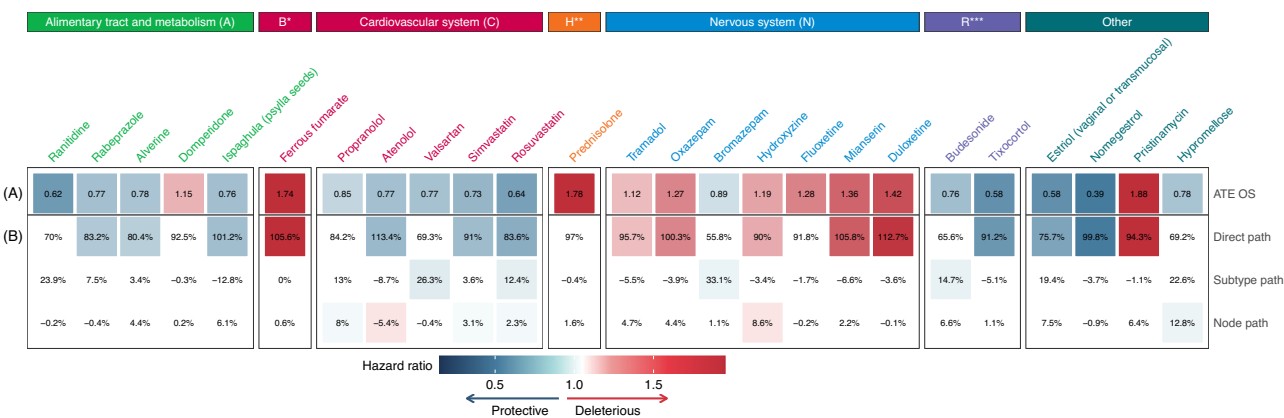

**Fig. 5 | Mediation analyzes for the 25 medications with a significant protective or deleterious effect for overall survival (OS).** Medications are grouped and color-coded by ATC level. Significant protective associations are shown in blue. Significant deleterious associations are shown in red. Non-significant associations are shown in white. **A** Average treatment effect (ATE) of the medication (Cox hazard ratio, HR). **B** Breakdown of the ATE into three path-specific effects (expressed as percentages): direct path, subtype path and node path. Source data are provided as a Source Data file. ATE average treatment effect, ATC Anatomical Therapeutic Chemical, HR hazard ratio, B* blood and blood-forming organs, H** systemic hormonal preparations, excluding sex hormones and insulins, R*** respiratory system.

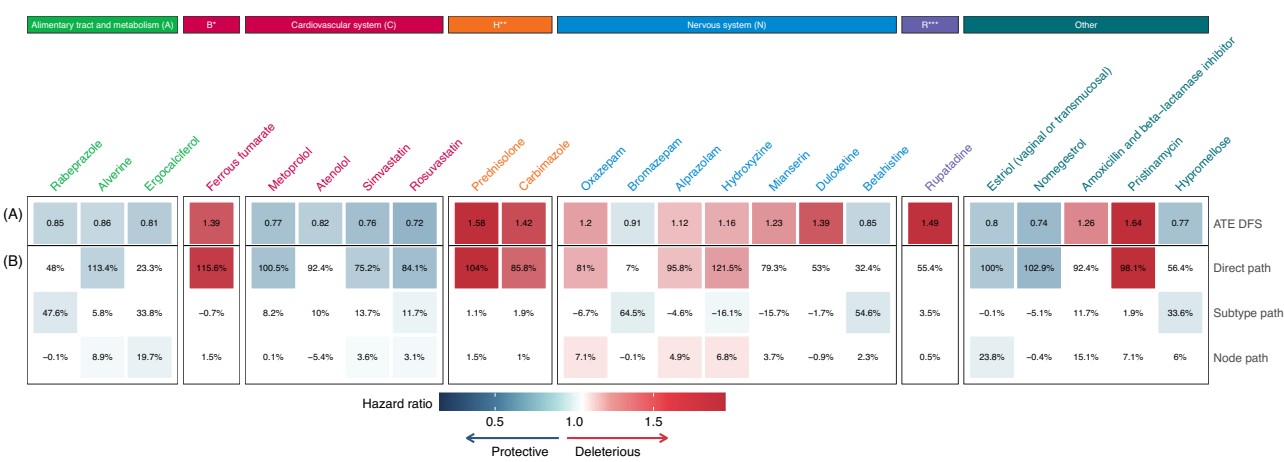

**Fig. 6 | Mediation analyzes for the 23 medications with a significant protective or deleterious effect for disease-free survival (DFS).** Medications are grouped and color-coded by ATC level. Significant protective associations are shown in blue. Significant deleterious associations are shown in red. Non-significant associations are shown in white. (**A**) Average treatment effect (ATE) of the medication (Cox hazard ratio, HR). (**B**) Breakdown of the ATE into three path-specific effects (expressed as percentages): direct path, subtype path, and node path. Source data are provided as a Source Data file. ATE average treatment effect, ATC Anatomical Therapeutic Chemical, HR hazard ratio, A* alimentary tract and metabolism, B** blood and blood-forming organs, H*** systemic hormonal preparations, excluding sex hormones and insulins, R**** respiratory system.

inclusion of comorbid conditions recorded up to 6 months after BC diagnosis allowed us to include conditions noted solely during BC hospitalizations, we could not rule out mismeasurement bias due to the inclusion of comorbid conditions triggered by BC diagnosis and treatment, such as depressive symptoms or postoperative acute phlebitis. However, in the two sensitivity analyzes we performed, our results were not affected by the timeframe chosen for the identification of comorbid conditions in hospital discharge reports, suggesting robustness of our study with respect to such measurement bias. Fourth, we could not overcome indication bias for some molecules (*e.g.* insulin), which were removed from the analyzes due to insufficient adjustment quality. Fifth, despite our high-dimensional adjustment and conservative quality check strategy, we cannot exclude the presence of unmeasured confounding bias for some molecules, which may weaken the causal interpretation of the results. These include lifestyle and behavioral factors such as diet, body mass index, smoking, alcohol consumption, or physical activity. While the presence of proxy indicators in our adjustment set (*e.g.*, severe obesity, diabetes, hypertension, severe tobacco

and alcohol dependence, deprivation index) may mitigate this limitation, we acknowledge the importance of direct measures of these variables in future research. Sixth, due to our limited follow-up period, with a median of four and a half years, our results relate predominantly to early deaths and recurrences. Finally, while our study provides foundational insights into the association between pre-diagnosis medication use and BC outcomes, we recognize that the absence of dose-response analysis limits the granularity of our findings. Future investigations are needed to determine the potential dose-dependent effects of pre-diagnosis medications on disease progression and patient survival.

Because our primary objective was to examine the effect of medication use prior to BC diagnosis in a population of patients with BC, our results do not reflect the effect of medication on BC risk, nor do they reflect the effect of medication use after cancer diagnosis or in a non-cancer population. Similarly, while we estimated the magnitude of the observed effect attributable to a change in BC subtype or lymph node involvement at diagnosis, we did not perform such mediation analysis for other BC biological characteristics, such as tumor stage,

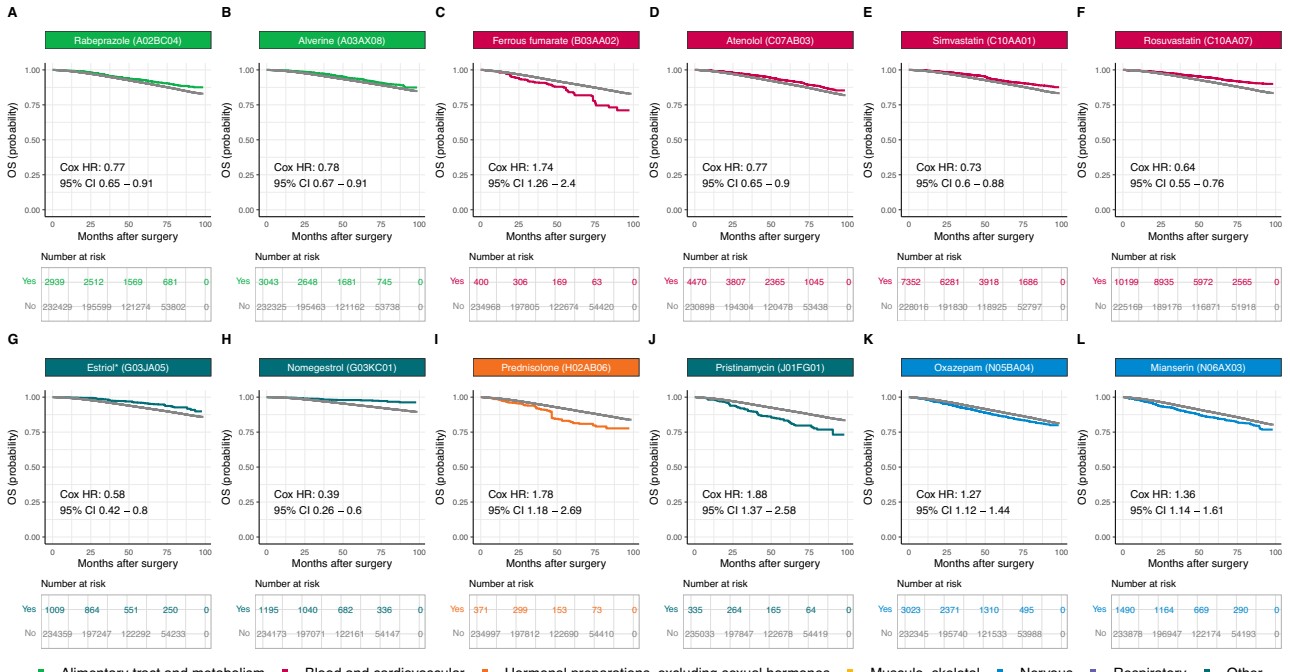

**Fig. 7 | Adjusted Kaplan-Meier survival curves for the 12 medications with a significant HR after adjustment for multiple testing for overall survival (OS).** **A** Rabeprazole (A02BC04); **B** Alverine (A03AX08); **C** Ferrous fumarate (B03AA02); **D** Atenolol (C07AB03); **E** Simvastatin (C10AA01); **F** Rosuvastatin (C10AA07); **G** Estriol* (vaginal or transmucosal) (G03JA05); **H** Nomegestrol (G03KC01); **I** Prednisolone (H02AB06); **J** Pristinamycin (J01FG01); **K** Oxazepam (N05BA04); **L** Mianserin (N06AX03). Survival curves for patients not on medication at the time of diagnosis are displayed in gray. Survival curves for patients with concomitant medication are color-coded by medication ATC level. The survival curves for all the other medications are available via the ADRENALINE web application (https://adrenaline.curie.fr/survival_analysis). Source data are provided as a Source Data file. * Vaginal or transmucosal. ATC Anatomical Therapeutic Chemical.

nor for post-diagnosis drug use, whose mediation portion remained included in the direct path-specific effect.

This work opens up new perspectives. From a research and development standpoint, academic or industrial researchers investigating molecules or pathways could confirm their hypotheses on human epidemiologic data and evaluate the magnitude of the effect on real-world evidence with this platform. This resource can also provide new hypotheses for drugs that may naturally influence BC evolution, from its presentation at diagnosis (subtype, lymph node involvement) to its long-term prognosis (overall survival, disease-free survival). The prospect of improving BC prognosis with affordable medications is particularly appealing, in a context in which the costs of innovative oncologic therapies could jeopardize healthcare systems. Hence, research to evaluate the effect of the drugs identified in this study after cancer diagnosis, including dose-response analyzes, is urgently needed.

## Methods
### Ethics and data protection
This study was conducted in the framework of a partnership between Institut Curie and INCa and was performed in accordance with institutional and ethical rules concerning research based on data from patients. The study was authorized by the French data protection agency (*Commission nationale de l'informatique et des libertés*−CNIL, under registration number 920017). In accordance with French regulations applicable to the SNDS, no informed consent was required.

### Data source and study population
We conducted a nationwide retrospective study with the published FRESH (French Early Breast Cancer Cohort) cohort[23]. The data released from the SNDS database available at the French National Cancer Institute (INCa)[20,69] which included (i) demographic data, (ii) hospital discharge reports, (iii) outpatient care, and (iv) long-term illness (LTI) records. The FRESH cohort includes all women with non-metastatic BC

newly diagnosed between January 1, 2011, and December 31, 2017, identified by a tag with a diagnosis code for BC in long-term illness records; or in at least one hospital discharge report within the period considered. The cohort excludes: (1) patients under the age of 18 years at inclusion (2) patients not affiliated to the principal national health insurance coverage plan ("Régime Général"), (3) patients not undergoing breast surgery in the year preceding or following inclusion, (4) patients with a concomitant cancer at another site, (5) patients with evidence of prior BC at diagnosis, (6) patients with distant metastases at BC diagnosis, and (7) patients with missing or inconsistent data (Supplementary Fig. 9). The date of BC diagnosis was taken as the date of either the earliest breast core biopsy in the year before the first breast surgery, or the earliest fine-needle aspiration cytology, or the earliest breast imaging procedure or the date of the first BC treatment. Details about the available data are provided in the Supplementary Methods.

### Concomitant medication
Concomitant medications were identified from outpatient drug delivery data the six months preceding BC diagnosis. Medications were classified according to the World Health Organization ATC (Anatomical Therapeutic Chemical) classification. We excluded diagnostic agents, medications used for cancer treatment and medications with no systemically active molecule (Supplementary Table 2). For medications based on combinations of molecules (e.g. beta-blocker and diuretics), we considered the individual components separately, with specific splitting rules applied for sex hormones (Supplementary Fig. 10). Chronic exposure to concomitant medication was coded as: (i) "yes" if the patient had received at least three months of the full dose in the six months preceding BC diagnosis; (ii) "no" otherwise. The decision rules for defining three months of full-dose treatment depended on the presentation and dose schedule of the medication, as described in the Supplementary Methods and Supplementary Table 3. In our analyzes, we focused specifically on chronic exposures to concomitant

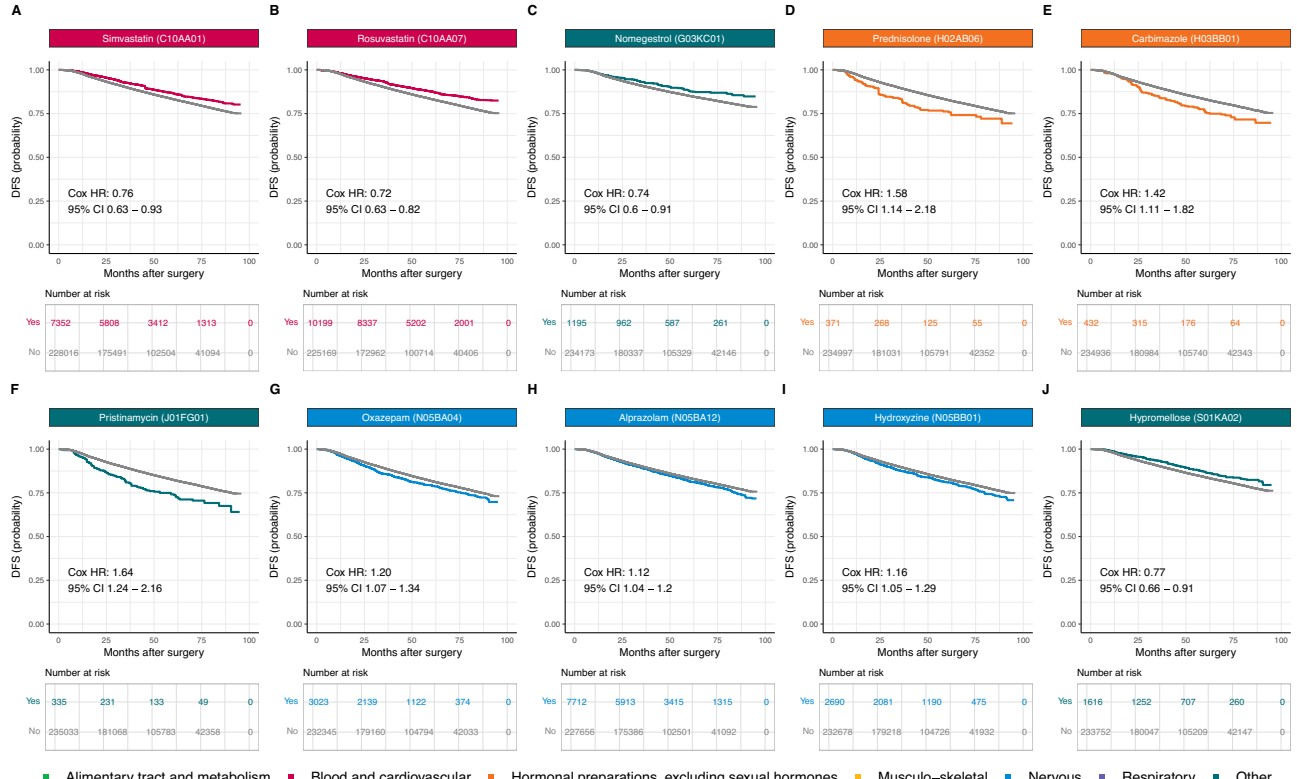

**Fig. 8 | Adjusted Kaplan-Meier survival curves for the 10 medications with a significant HR after adjustment for multiple testing for disease-free survival (DFS). A** Simvastatin (C10AA01); **B** Rosuvastatin (C10AA07); **C** Nomegestrol (G03KC01); **D** Prednisolone (H02AB06); **E** Carbimazole (H03BB01); **F** Pristinamycin (J01FG01); **G** Oxazepam (N05BA04); **H** Alprazolam (N05BA12); **I** Hydroxyzine (N05BB01); **J** Hypromellose (S01KA02). Survival curves for patients not on medication at the time of diagnosis are displayed in gray. Survival curves for patients with concomitant medication are color-coded by medication ATC level. The survival curves for all the other medications are available via the ADRENALINE web application (https://adrenaline.curie.fr/survival_analysis). Source data are provided as a Source Data file. ATC Anatomical Therapeutic Chemical.

medication. We restricted the analysis to medications taken by at least 300 patients, to ensure sufficient statistical power. The reference group for comparison included both patients who were never exposed to the medication and patients who had non-chronic exposure to this medication.

## Comorbid conditions

In total, we identified 52 comorbid conditions, belonging to 12 categories (Supplementary Data 4), as proposed in previous studies[25–34]. The presence of a disease at BC diagnosis was detected on the basis of procedure codes in the year before BC diagnosis up to BC diagnosis, and of diagnosis codes in the year before BC diagnosis up to 180 days after BC diagnosis. We used diagnosis codes up to 180 days after diagnosis to include the comorbid conditions noted by the surgeon at the time of first surgery for BC. As sensitivity analyzes, we tested two additional timeframes for diagnosis codes: (i) the year prior to BC diagnosis up to BC surgery, and (ii) the one-year period prior to BC surgery. Further details are provided in the Supplementary Methods.

## Other covariates

Other covariates were split into (i) pre-exposure and (ii) post-exposure covariates. Pre-exposure covariates included: (1) age at BC diagnosis, (2) the deprivation index of the area of residence[70], (3) the number of general practitioner (GP) visits in the year preceding BC diagnosis, (4) the number of visits to a gynecologist in the year preceding BC diagnosis, (5) the performance of a mammographic screening in the year preceding BC diagnosis, (6) the total number of medications (molecules) to which the patient was chronically exposed to the six months preceding BC diagnosis, and (7) concomitant exposure to other medications. Post-exposure covariates included: (1) BC subtype, (2) nodal status, (3) chemotherapy status, and (4) endocrine therapy status. Further details are provided in the Supplementary Methods.

## Outcomes

The primary endpoint was overall survival (OS). Disease-free survival (DFS) was evaluated as a secondary endpoint. OS was defined as the time, in months, from the first BC surgery to death or to March 1, 2019, whichever occurred first. Vital status and date of deaths were directly available in the SNDS data. DFS was defined as the time, in months, from the first BC surgery to death, loco-regional recurrence, contralateral recurrence, distant recurrence, or 30th of December 2018, whichever occurred first. Of note, we did not include the second cancer of another site (non-breast) in the definition. The occurrence of any of loco-regional recurrence, distant recurrence, or contralateral recurrence, was identified based on (i) the resumption of radiotherapy, chemotherapy, or targeted therapy at least 6 months after the end of the initial treatments, (ii) a breast surgery procedure with axillar procedure performed at least 6 months after the end of the initial treatments, (iii) the intake of an anti-cancer molecule approved only in the metastatic setting starting at least six months after initial breast surgery, or (iv) the presence of a diagnosis code of metastasis in hospitalization stays starting at least six months after initial breast surgery (Supplementary Data 5). Breast surgery was tagged with hospital procedure codes for mastectomy and partial mastectomy.

## Causal inference pipeline

A directed acyclic graph representing the expected causal links between variables was built in accordance with expert knowledge

(Supplementary Fig. 11)[71], and was used to identify the pre-exposure covariates that needed to be adjusted on in the analyzes. Our goal was to estimate the average effect of each medication in the entire population (average treatment effect, ATE)[71]. The causal inference pipeline (Fig. 9) was run for each medication, one at a time. It could be broken down into five steps. Further details on the methods used are provided in the Supplementary Methods.

**Step 1: Adjustment by inverse probability of treatment weighting (IPTW).** We used inverse probability of treatment weighting (IPTW) to adjust for the confounding bias induced by pre-exposure covariates identified as confounding factors in the DAG analysis. This procedure involved: (i) estimating propensity scores (PS) *i.e.* the probability of receiving the drug concerned given the value of the pre-exposure covariates for each patient; (ii) weighting the dataset by

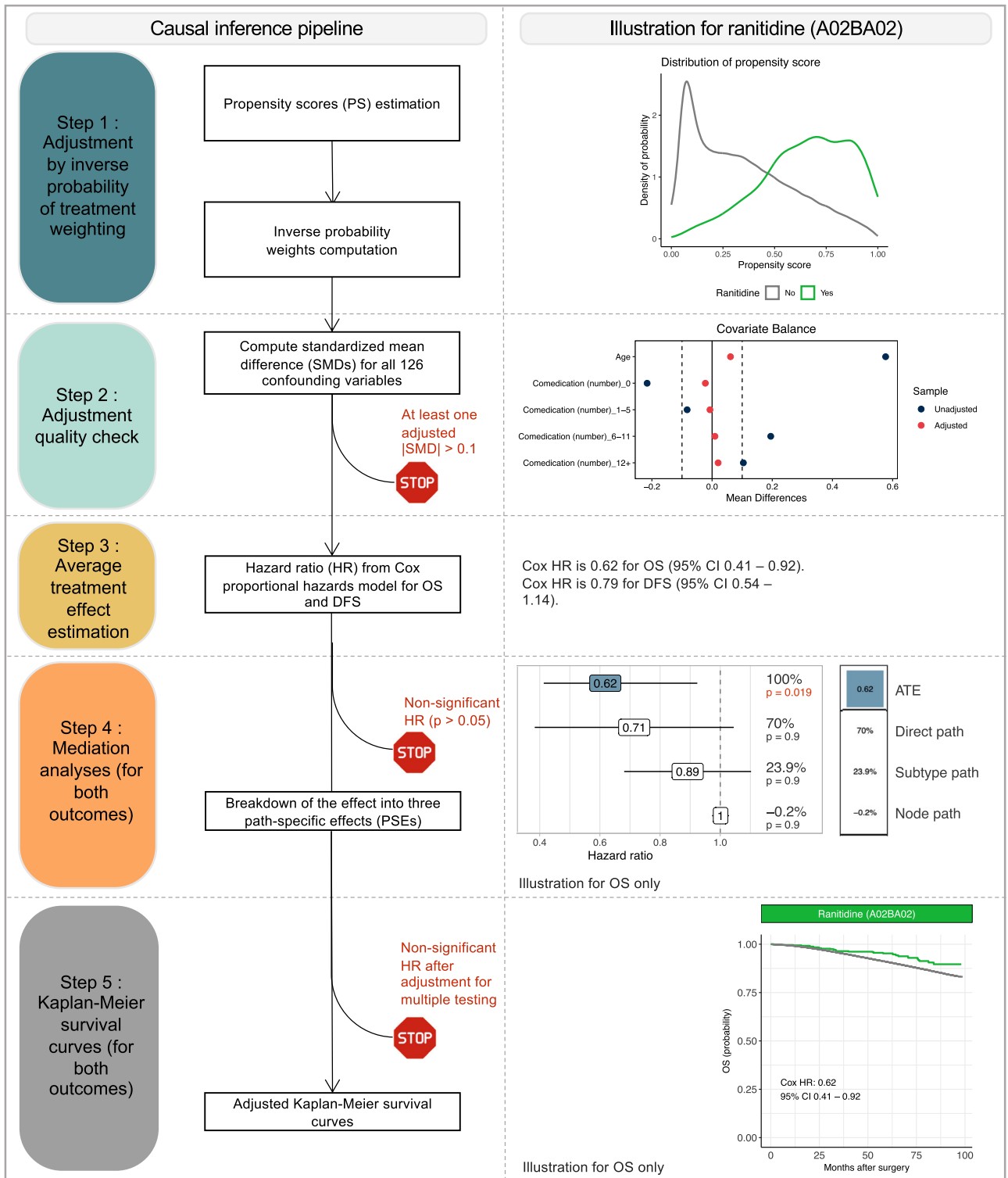

**Fig. 9 | Causal inference pipeline of the study (left) and illustration for one medication (ranitidine, ATC code A02BA02, right).** Details are provided in the Methods. Abbreviations: PS propensity score, IPTW inverse probability of treatment weighting, SMD standardized mean difference, ATE average treatment effect, HR hazard ratio, ATC Anatomical Therapeutic Chemical, CT chemotherapy, TNBC triple-negative breast cancer, OS overall survival, DFS disease-free survival.

assigning each patient a stabilized inverse probability weight derived from the PS. All subsequent analyzes were conducted on the weighted dataset.

**Step 2: Adjustment quality check.** We checked the adjustment quality *a posteriori* by calculating the standardized mean differences (SMDs) for each pre-exposure covariate after adjustment. In accordance with published results[72], the adjustment quality was considered insufficient if any SMD had an absolute value above 0.1, in which case the molecule was discarded from subsequent analyzes.

**Step 3: Average treatment effect estimation.** We estimated the average treatment effect (ATE) by calculating the hazard ratio (HR) of a univariate Cox proportional hazards model fitted to the weighted population[73]. We used Wald tests calculated with robust covariances to draw statistical inferences about the estimated HR. The threshold for statistical significance was $p = 0.05$.

**Step 4: Mediation analyzes.** Molecules with a significant ATE were selected for mediation analyzes, which involved breaking down the ATE into several pathways passing through two potential mediators, BC subtype and nodal status (Supplementary Fig. 12). We assumed BC subtype and nodal status to be causally related. Standard direct and indirect effects were not, therefore, directly identifiable for each mediator[74]. It was, nevertheless, possible to break the ATE down into three path-specific effects (PSEs): (1) the effect through pathways involving neither a difference in BC subtype nor in nodal status (direct effects); (2) the effect through pathways involving a difference in nodal status only (effect through node); (3) the effect through pathways involving a difference in BC subtype (and potentially involving a difference in nodal status; effect through subtype). PSEs were estimated by a weighting approach[74] and are expressed as percentages of the total effect (which may be negative).

**Step 5: Kaplan–Meier survival curves.** Weighted Kaplan-Meier survival curves were plotted for the molecules with a significant ATE after the Benjamini-Hochberg (BH) multiple testing procedure, and compared with an adjusted log-rank test[75]. The threshold for statistical significance was set at $p = 0.1$, due to the low power of adjusted log-rank tests[76].

### Web application and software
All the results are available via an interactive web application (https://adrenaline.curie.fr), also including: (i) a comprehensive descriptive overview of the database; (ii) the results of the causal inference pipeline for medication classes (ATC levels 2, 3, and 5), and for medications failing the adjustment quality test; and (iii) subgroup analyzes by BC subtype, nodal status, age, chemotherapy status, and endocrine therapy status. Analyzes were performed with R software, version 3.6.3 (see Supplementary Methods for details). All hypothesis tests were two-tailed.

### Reporting summary
Further information on research design is available in the Nature Portfolio Reporting Summary linked to this article.

## Data availability
The raw SNDS data are protected and are not available due to data privacy laws. The processed aggregated data generated in this study are provided in the Supplementary Information/Source Data file. Source data for the Figures are provided with this paper. Source data are provided with this paper.

## Code availability
Code is available online (https://doi.org/10.5281/zenodo.10777521)[77].

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

## Acknowledgements

We thank the Department of Health Data and Assessment, Health Survey Data Science and Assessment Division, French National Cancer Institute (Institut National du Cancer INCa) for providing us with access to the cancer cohort. E.D. received a PhD grant from the Ministère de l'Enseignement Supérieur et de la Recherche et de l'Innovation, allocated to École polytechnique (AMX). This study was also funded by Monoprix* (A.S.H. and F.R.) and INCa grant number 18–127 within the COMBIMMUNO (Comedications and comorbidities in breast cancer: Deciphering Interactions Between Immune Infiltration, Response to Treatment and Prognosis) project (A.S.H. and F.R.). This work was supported by an Investissement d'Avenir grant from Agence Nationale de la Recherche, reference ANR-19-P3IA-0001 (PRAIRIE 3IA Institute) (C.A.A.). The funder was not involved in study design, or in the collection, analysis, and interpretation of data, the writing of this article or the decision to submit it for publication.

## Author contributions

A.S.H. and F.R. conceived the project and designed the study. E.D. carried out the statistical analysis and prepared the figures, tables, and interactive online platform. B.M., A.B., E.D.N., T.D. and S.A. participated in discussions on analysis design and refinement. C.A.A., J.A., B.A. and A.L. revised the statistical analysis and the causal framework. S.H., C.L.B.B. and P.J.B. ensured data availability and quality and participated in the design of data preparation. B.G.R., P.G., F.J., M.E., E.L., F.C., J.Y.P. and J.H. provided medical background knowledge for data identification and analysis. The manuscript was drafted by E.D., A.S.H. and C.B. and edited and approved by all authors.

## Competing interests

The authors declare no competing interests.

## Additional information

[1]Residual Tumor & Response to Treatment Laboratory, RT2Lab, Translational Research Department, INSERM, U932 Immunity and Cancer, Université Paris Cité, F-75005 Paris, France. [2]INSERM, U900, 75005 Paris, France. [3]MINES ParisTech, PSL Research University, CBIO-Centre for Computational Biology, 75006 Paris, France. [4]Health Data and Assessment, Health Survey Data Science and Assessment Division, French National Cancer Institute (Institut National du Cancer INCa), 92100 Boulogne-Billancourt, France. [5]INRIA, Paris-Saclay University, CEA, Palaiseau 91120, France. [6]Department of Gynecology, Strasbourg University Hospital, Strasbourg, France. [7]INSERM UMR-S 900, Institut Curie, MINES ParisTech CBIO, PSL Research University, 92210 Saint-Cloud, France. [8]Département de Recherche Translationnelle - Plateforme Biophenics, PICT-IBISA, PSL Research University, Paris, France. [9]Institut Curie - PSL Research University Translational Research Department Breast Cancer Biology Group 26 rue d'Ulm, 75005 Paris, France. [10]Institut Curie, PSL Research University, Uveal Melanoma Group, Translational Research Department, Paris, France. [11]Department of Biostatistics, Unicancer, Paris, France. [12]Conservatoire National des Arts et Métiers, Paris, France. [13]Breast diseases Center Hôpital saint Louis APHP, Université Paris Cité, Paris, France. [14]Department of Surgical Oncology, Université Paris Cité, Institut Curie, 75005 Paris, France. [15]Department of Medical Oncology, Université Paris Cité, Institut Curie, 75005 Paris, France. [16]Aix Marseille Univ, Inserm, IRD, SESSTIM, Équipe Labellisée Ligue Contre le Cancer, 13005 Marseille, France. [17]Health Survey Data Science and Assessment Division, French National Cancer Institute (Institut National du Cancer INCa), 92100 Boulogne-Billancourt, France. [18]Department of Surgery, Institut Jean Godinot, Reims, France. [19]Institut Curie, PSL Research University, Paris, France. ✉e-mail: fabien.reyal@curie.fr

