## [Peer Review File · Nature Communications]

REVIEWER COMMENTS

Reviewer #1 (Remarks to the Author):

The revised manuscript is much improved. Thank you for correcting the terms and adding additional analysis.

Reviewer #2 (Remarks to the Author):

I thank the authors for their detailed answers to my comments / questions, that I found overall very satisfactory.

Please find below a few remarks following their answers:

- “Web application and software” (pages 19-20): please specify here that subgroup analyses by age are also available.

- I would like to comment on their answer pertaining to the fact that comorbid conditions were identified using (among other sources) hospital discharge data pertaining to the 1-year period preceding BC up to 180 days after the date of BC diagnosis, which means in many cases after the first BC surgery. Using data collected after the start of follow-up (BC surgery) to classify individuals from the start of follow-up onwards can lead to biased estimates. Although the consequences should be less dramatic when such misclassification concerns adjustment factors rather than exposure data [doi: 10.1002/pds.5083], I think that the issue deserves some consideration. I understand the author’s arguments for using information from hospital discharge reports up to 180 days after the date of BC diagnosis rather than up to BC surgery. However, it is important to see what the estimated ATE would be if hospital discharge reports had been used for the identification of comorbidities only up to BC surgery. This could simply be a sensitivity analysis, run only for the 16 medications identified as associated with either DFS or OS or both (rather than on the entire causal inference pipeline!). An alternative way of identifying/adjust for comorbidities would be to use data other than hospital discharge pertaining to the 1-year period preceding BC diagnosis and hospital discharge reports pertaining to the 1-year period preceding BC surgery (+ the report pertaining to BC surgery). This could also be investigated in a simple sensitivity analysis, just to reassure the reader that ATE estimates are not much affected.

- In the Discussion section, the authors state that “Sex hormones have previously been associated with an increased risk of BC” and cite references #48 and #49 in support of that sentence. However, reference #48 is about the use of hormone therapy in cancer survivors, and reference #49 does not focus on breast cancer incidence. I think that other references would be more appropriate, such as the one from the Collaborative Group on Hormonal Factors in Breast Cancer published in 2019 [doi: 10.1016/S0140-

6736(19)31709-X] regarding observational data, and one or two references related to the WHI randomized controlled trial.

- In the discussion section, the authors state: “Our results on vaginal estrogens are also consistent with previous findings of their association with a lower risk of all-cause mortality when used after BC” and cite references #50, 51 and 52 to support the sentence. However, in ref #50, exposure to vaginal estrogens is assessed before breast cancer occurrence, not after; and in ref #52, the outcome is not all-cause mortality but breast cancer mortality.

- When discussing their results about prednisolone, the authors state: “Our study revealed that prednisolone was associated with a 58% decrease in DFS and a 78% decrease in OS, in line with several preclinical studies suggesting that glucocorticoids may promote BC progression and metastasis (59–61)”. Besides preclinical studies, a recent epidemiological study is also in line with these results [DOI: 10.1186/s12916-021-02004-6]: in that cohort study, the use of glucocorticoids was associated with decreased risk of stage 1 or stage 2 tumours but increased risk of stage 3 or 4 cancers.

Reviewer #5 (commented on the authors' response to original Reviewers #3 and #4 (Remarks to the Author):

As requested I have reviewed the degree to which the authors have responded to the comments from referees #3 and #4.

Reviewer 3's main concern was that the study was based upon medications used before diagnosis to infer the impact of medications after diagnosis on breast cancer-specific mortality. I think reviewer 3 is making an important point. In response the authors removed and added text to clarify that pre-diagnosis medications were being investigated e.g. “nor do they reflect the effect of medication use after cancer diagnosis or in a non-cancer population.” They also changed the conclusion to: “This resource can guide the choice of candidates for drug repurposing trials or pharmacovigilance warnings.” This still seems to apply to me that they are inferring the effect of the medication before diagnosis will be the same as after diagnosis. For instance, you cannot change the medications you have been using before your breast cancer diagnosis – so it is hard to see this as useful information for pharmacovigilance. And surely repurposing trials should be informed by medication use after breast cancer diagnosis.

Reviewer 3 was also concerned about the lack of analysis of dose response and the use of a binary 3 month use in 6 month definition. I think I would agree a dose response analysis is a non-trivial omission and dose response analyses are common in this kind of screening study (e.g. doi: 10.1016/j.ebiom.2016.03.018.). The authors have at least added this as a limitation.

Reviewer 3 was also concerned about the use of all-cause mortality rather than breast cancer-specific mortality. Here the authors have been responsive and have added disease free survival but not breast cancer specific mortality. They now mention this as a “main outcome”. I am not sure changing the main outcome is appropriate or transparent as their main outcome should be the main outcome of their original proposal.

Reviewer 3 discussed missing confounders including stage. The authors argue it is not a confounder but a mediator – which seems a reasonable argument. The authors though focus part of their analysis on mediation so missing stage remains an important omission. Surely if any of these medications impacted outcomes it could do so through stage and if mediation was an important goal of the study you would want to investigate mediation by stage. In general, they do not discuss other potential confounders in any real detail. For instance, BMI, smoking, alcohol intake, physical activity etc.

Reviewer 3 was concerned that the figures were hard to read. I would have to agree and I don't think the figures have changed much.

I would agree with many of Reviewer 4's concerns. The cohort is poorly described – and this has not improved much e.g. I could not find a simple table of breast cancer characteristics anywhere – this is a strange omission. Also I agree that the methods are (overly) complicated and not well described and this has not improved much. I think if you are going to publish this, an expert in causal analysis should thoroughly review their methods.

Unrelated to the comments of the reviewers, a further weakness, not discussed, is that the cohort has quite short follow-up for mortality (of <5 years). Breast cancer can recur many years after diagnosis and I think throughout the authors should really refer to early death/recurrence.

The authors suggest estrogen (vaginal/transmucosal) is protective in breast cancer patients but they should really address the randomised controlled trials of estrogen (<https://doi.org/10.1093/jnci/djs014>) and tibolone (Lancet Oncol. 2009 Feb;10(2):135-46) which have shown increases in breast cancer recurrence in breast cancer patients using either medication.

Reviewer #2:

I thank the authors for their detailed answers to my comments / questions, that I found overall very satisfactory. Please find below a few remarks following their answers:

We thank reviewer #2 for taking the time to review our responses. We have specifically addressed each of the additional comments/questions below.

- 1. “Web application and software” (pages 19-20): please specify here that subgroup analyses by age are also available.**

We thank the reviewer for the suggestion and apologize for the oversight. We have revised the manuscript as follows:

Page 22, line 577,

“(iii) subgroup analyses by BC subtype, nodal status, chemotherapy status, and endocrine therapy status.”

was replaced by

*“(iii) subgroup analyses by BC subtype, nodal status, **age**, chemotherapy status, and endocrine therapy status.”*

- 2. I would like to comment on their answer pertaining to the fact that comorbid conditions were identified using (among other sources) hospital discharge data pertaining to the 1-year period preceding BC up to 180 days after the date of BC diagnosis, which means in many cases after the first BC surgery. Using data collected after the start of follow-up (BC surgery) to classify individuals from the start of follow-up onwards can lead to biased estimates. Although the consequences should be less dramatic when such misclassification concerns adjustment factors rather than exposure data [doi: 10.1002/pds.5083], I think that the issue deserves some consideration. I understand the author’s arguments for using information from hospital discharge reports up to 180 days after the date of BC diagnosis rather than up to BC surgery. However, it is important to see what the estimated ATE would be if hospital discharge reports had been used for the identification of comorbidities only up to BC surgery. This could simply be a sensitivity analysis, run only for the 16 medications identified as associated with either DFS or OS or both (rather than on the entire causal inference pipeline!). [...]**

We thank the reviewer for this suggestion and recognize the importance of ensuring that our classification of comorbid conditions does not inadvertently introduce bias into our estimation of the average treatment effect.

As suggested by the reviewer, we performed a sensitivity analysis focusing only on the 16 drugs identified as being associated with either disease-free survival (DFS), overall survival (OS), or both. For this analysis, we restricted the identification of comorbid conditions to (i) non-hospital discharge data pertaining to the 1-year period prior to BC diagnosis and (ii) hospital discharge data pertaining to the 1-year period prior to BC diagnosis through the date of BC surgery (included).

The results of this sensitivity analysis are shown in Table R1, along with initial results. The effects estimated with this alternative definition of comorbid conditions were almost identical to the original results (with identification of comorbidities up to 180 days after BC diagnosis), with no change in the direction and the significance of the estimates, and very little change in the magnitude.

ATC code	Molecule	Disease-free survival						Overall survival					
		Hospital discharge reports up to 180 days after BC diagnosis (initial definition)			Hospital discharge reports up to BC surgery (sensitivity analysis 1)			Hospital discharge reports up to 180 days after BC diagnosis (initial definition)			Hospital discharge reports up to BC surgery (sensitivity analysis 1)		
		HR	95% CI	P	HR	95% CI	P	HR	95% CI	P	HR	95% CI	P
A02BC04	Rabeprazole	0.85	0.72 - 0.99	.04	0.84	0.72 - 0.99	.03	0.77	0.65 - 0.91	.002	0.77	0.65 - 0.91	.002
A03AX08	Alverine	0.86	0.76 - 0.96	.008	0.86	0.77 - 0.97	.01	0.78	0.67 - 0.91	.002	0.79	0.67 - 0.92	.002
B03AA02	Ferrous fumarate	1.39	1.07 - 1.81	.01	1.38	1.07 - 1.78	.01	1.74	1.26 - 2.4	<.001	1.71	1.25 - 2.33	<.001
C07AB03	Atenolol	0.82	0.7 - 0.97	.02	0.80	0.66 - 0.95	.01	0.77	0.65 - 0.9	.001	0.74	0.62 - 0.88	<.001
C10AA01	Simvastatin	0.76	0.63 - 0.93	.006	0.76	0.63 - 0.92	.006	0.73	0.6 - 0.88	.001	0.73	0.61 - 0.88	.001
C10AA07	Rosuvastatin	0.72	0.63 - 0.82	<.001	0.72	0.63 - 0.82	<.001	0.64	0.55 - 0.76	<.001	0.64	0.55 - 0.75	<.001
G03JA05	Estriol (vaginal or transmucosal)	0.80	0.64 - 0.99	.04	0.77	0.62 - 0.96	.02	0.58	0.42 - 0.8	<.001	0.55	0.4 - 0.75	<.001
G03KC01	Nomegestrol	0.74	0.6 - 0.91	.005	0.73	0.59 - 0.9	.003	0.39	0.26 - 0.6	<.001	0.40	0.26 - 0.61	<.001
H02AB06	Prednisolone	1.58	1.14 - 2.18	.005	1.65	1.19 - 2.29	.002	1.78	1.18 - 2.69	.006	1.77	1.18 - 2.66	.006
H03BB01	Carbimazole	1.42	1.11 - 1.82	.005	1.45	1.14 - 1.85	.003	1.25	0.93 - 1.68	.14	1.27	0.95 - 1.7	.1
J01FG01	Pristinamycin	1.64	1.24 - 2.16	<.001	1.65	1.25 - 2.17	<.001	1.88	1.37 - 2.58	<.001	1.90	1.39 - 2.61	<.001
N05BA04	Oxazepam	1.20	1.07 - 1.34	.002	1.22	1.09 - 1.36	<.001	1.27	1.12 - 1.44	<.001	1.30	1.15 - 1.47	<.001
N05BA12	Alprazolam	1.12	1.04 - 1.2	.003	1.12	1.04 - 1.2	.002	1.06	0.97 - 1.17	.21	1.07	0.98 - 1.18	.14
N05BB01	Hydroxyzine	1.16	1.05 - 1.29	.005	1.18	1.06 - 1.31	.002	1.19	1.04 - 1.35	.009	1.21	1.07 - 1.38	.003
N06AX03	Mianserin	1.23	1.05 - 1.44	.01	1.23	1.05 - 1.44	.01	1.36	1.14 - 1.61	<.001	1.36	1.14 - 1.62	<.001
S01KA02	Hypromellose	0.77	0.66 - 0.91	.002	0.77	0.65 - 0.91	.002	0.78	0.63 - 0.96	.02	0.77	0.62 - 0.94	.01

Table R1: Comparison of initial results where comorbid conditions were identified from (i) non-hospital discharge data pertaining to the 1-year period prior to BC diagnosis and (ii) hospital discharge data pertaining to the 1-year period prior to BC diagnosis **through 180 days post BC diagnosis**, and results of a **first sensitivity analysis**, where comorbid conditions were identified from (i) non-hospital discharge data pertaining to the 1-year period prior to BC diagnosis and (ii) hospital discharge data pertaining to the 1-year period prior to BC diagnosis **through the date of BC surgery**. Abbreviations: HR: Hazard Ratio; CI: Confidence Interval.

[...] An alternative way of identifying/adjust for comorbidities would be to use data other than hospital discharge pertaining to the 1-year period preceding BC diagnosis and hospital discharge reports pertaining to the 1-year period preceding BC surgery (+ the report pertaining to BC surgery). This could also be investigated in a simple sensitivity analysis, just to reassure the reader that ATE estimates are not much affected.

We thank the reviewer for this alternative suggestion. As proposed, we performed a second sensitivity analysis for the sixteen molecules associated with DFS, OS, or both, but restricting the identification of comorbidities to (i) non-hospital discharge data pertaining to the 1-year period prior to BC diagnosis and (ii) hospital discharge data pertaining **to the 1-year period prior to BC surgery.**

The results of this sensitivity analysis are shown in Table R2. Again, ATE estimates were not much affected, with notably no change in significance or direction of the estimates.

⇒ **Overall, these additional analyses support the validity and robustness of our original findings, in which comorbidities were defined based on hospital discharge data up to 180 days after the diagnosis of BC. As recommended by the reviewer, we have included the results of these two sensitivity analyses in the revised manuscript, as follows.**

We added a supplementary table (Supplementary Table 1, displayed here as Table R3), which contains the results of the two sensitivity analyses for the sixteen molecules associated with either OS, DFS, or both.

In the supplementary material, page 6, line 122

“Sensitivity analyses were conducted to evaluate the impact of comorbid conditions identification timing on the results. Two additional timeframes for diagnosis codes in hospital discharge reports were tested: (i) the year before BC diagnosis up to BC surgery, and (ii) the one-year period before BC surgery; resulting in two alternative definitions of comorbid conditions. The causal inference pipeline was then re-run with the modified definitions of comorbid conditions for the sixteen molecules that were identified as being associated with either OS, DFS, or both.”

was added.

In the main manuscript, Methods section, page 19, line 488

“As sensitivity analyses, we tested two additional timeframes for diagnosis codes: (i) the year prior to BC diagnosis up to BC surgery, and (ii) the one-year period prior to BC surgery.”

was added.

In the main manuscript, results section, page 9, line 244

“We obtained similar results in sensitivity analyses performed with two different timeframes for the identification of comorbid conditions (Supplementary Table 1).”

was added.

In the main manuscript, discussion section, page 15, line 393

“However, in the two sensitivity analyses we performed, our results were not affected by the timeframe chosen for the identification of comorbid conditions in hospital discharge reports, suggesting robustness of our study with respect to such measurement bias.”

was added.

ATC code	Molecule	Disease-free survival						Overall survival					
		Hospital discharge reports up to 180 days after BC diagnosis (initial definition)			Hospital discharge reports pertaining to the 1-year period prior to BC surgery (sensitivity analysis 2)			Hospital discharge reports up to 180 days after BC diagnosis (initial definition)			Hospital discharge reports pertaining to the 1-year period prior to BC surgery (sensitivity analysis 2)		
		HR	95% CI	P	HR	95% CI	P	HR	95% CI	P	HR	95% CI	P
A02BC04	Rabeprazole	0.85	0.72 - 0.99	.04	0.84	0.72 - 0.99	.04	0.77	0.65 - 0.91	.002	0.77	0.65 - 0.91	.002
A03AX08	Alverine	0.86	0.76 - 0.96	.008	0.86	0.77 - 0.97	.01	0.78	0.67 - 0.91	.002	0.79	0.67 - 0.92	.002
B03AA02	Ferrous fumarate	1.39	1.07 - 1.81	.01	1.37	1.06 - 1.77	.02	1.74	1.26 - 2.4	<.001	1.70	1.25 - 2.32	<.001
C07AB03	Atenolol	0.82	0.7 - 0.97	.02	0.79	0.66 - 0.95	.01	0.77	0.65 - 0.9	.001	0.74	0.63 - 0.88	<.001
C10AA01	Simvastatin	0.76	0.63 - 0.93	.006	0.76	0.63 - 0.92	.005	0.73	0.6 - 0.88	.001	0.73	0.6 - 0.88	<.001
C10AA07	Rosuvastatin	0.72	0.63 - 0.82	<.001	0.72	0.63 - 0.82	<.001	0.64	0.55 - 0.76	<.001	0.64	0.55 - 0.75	<.001
G03JA05	Estriol (vaginal or transmucosal)	0.80	0.64 - 0.99	.04	0.77	0.62 - 0.96	.02	0.58	0.42 - 0.8	<.001	0.55	0.4 - 0.75	<.001
G03KC01	Nomegestrol	0.74	0.6 - 0.91	.005	0.73	0.59 - 0.91	.005	0.39	0.26 - 0.6	<.001	0.40	0.26 - 0.62	<.001
H02AB06	Prednisolone	1.58	1.14 - 2.18	.005	1.65	1.19 - 2.29	.003	1.78	1.18 - 2.69	.006	1.77	1.18 - 2.66	.006
H03BB01	Carbimazole	1.42	1.11 - 1.82	.005	1.45	1.14 - 1.85	.003	1.25	0.93 - 1.68	.14	1.27	0.95 - 1.7	.1
J01FG01	Pristinamycin	1.64	1.24 - 2.16	<.001	1.64	1.25 - 2.16	<.001	1.88	1.37 - 2.58	<.001	1.90	1.38 - 2.6	<.001
N05BA04	Oxazepam	1.20	1.07 - 1.34	.002	1.22	1.09 - 1.36	<.001	1.27	1.12 - 1.44	<.001	1.30	1.15 - 1.48	<.001
N05BA12	Alprazolam	1.12	1.04 - 1.2	.003	1.12	1.04 - 1.2	.002	1.06	0.97 - 1.17	.21	1.07	0.97 - 1.18	.16
N05BB01	Hydroxyzine	1.16	1.05 - 1.29	.005	1.18	1.06 - 1.31	.003	1.19	1.04 - 1.35	.009	1.21	1.07 - 1.38	.003
N06AX03	Mianserin	1.23	1.05 - 1.44	.01	1.23	1.05 - 1.44	.01	1.36	1.14 - 1.61	<.001	1.35	1.14 - 1.61	<.001
S01KA02	Hypromellose	0.77	0.66 - 0.91	.002	0.77	0.65 - 0.91	.002	0.78	0.63 - 0.96	.02	0.77	0.62 - 0.94	.01

Table R2: Comparison of initial results where comorbid conditions were identified from (i) non-hospital discharge data pertaining to the 1-year period prior to BC diagnosis and (ii) hospital discharge data pertaining to the 1-year period prior to BC diagnosis **through 180 days post BC diagnosis**, and results of a **second sensitivity analysis**, where comorbid conditions were identified from (i) non-hospital discharge data pertaining to the 1-year period prior to BC diagnosis and (ii) hospital discharge data pertaining **the 1-year period prior to BC surgery**. Abbreviations: *HR*: Hazard Ratio; *CI*: Confidence Interval.

ATC code	Molecule	Overall survival						Disease-free survival					
		Hospital discharge reports up to BC surgery (sensitivity analysis 1)			Hospital discharge reports pertaining to the 1-year period prior to BC surgery (sensitivity analysis 2)			Hospital discharge reports up to BC surgery (sensitivity analysis 1)			Hospital discharge reports pertaining to the 1-year period prior to BC surgery (sensitivity analysis 2)		
		HR	95% CI	P	HR	95% CI	P	HR	95% CI	P	HR	95% CI	P
A02BC04	Rabeprazole	0.77	0.65 - 0.91	.002	0.77	0.65 - 0.91	.002	0.84	0.72 - 0.99	.03	0.84	0.72 - 0.99	.04
A03AX08	Alverine	0.79	0.67 - 0.92	.002	0.79	0.67 - 0.92	.002	0.86	0.77 - 0.97	.01	0.86	0.77 - 0.97	.01
B03AA02	Ferrous fumarate	1.71	1.25 - 2.33	<.001	1.70	1.25 - 2.32	<.001	1.38	1.07 - 1.78	.01	1.37	1.06 - 1.77	.02
C07AB03	Atenolol	0.74	0.62 - 0.88	<.001	0.74	0.63 - 0.88	<.001	0.80	0.66 - 0.95	.01	0.79	0.66 - 0.95	.01
C10AA01	Simvastatin	0.73	0.61 - 0.88	.001	0.73	0.6 - 0.88	<.001	0.76	0.63 - 0.92	.006	0.76	0.63 - 0.92	.005
C10AA07	Rosuvastatin	0.64	0.55 - 0.75	<.001	0.64	0.55 - 0.75	<.001	0.72	0.63 - 0.82	<.001	0.72	0.63 - 0.82	<.001
G03JA05	Estriol (vaginal or transmucosal)	0.55	0.4 - 0.75	<.001	0.55	0.4 - 0.75	<.001	0.77	0.62 - 0.96	.02	0.77	0.62 - 0.96	.02
G03KC01	Nomegestrol	0.40	0.26 - 0.61	<.001	0.40	0.26 - 0.62	<.001	0.73	0.59 - 0.9	.003	0.73	0.59 - 0.91	.005
H02AB06	Prednisolone	1.77	1.18 - 2.66	.006	1.77	1.18 - 2.66	.006	1.65	1.19 - 2.29	.002	1.65	1.19 - 2.29	.003
H03BB01	Carbimazole	1.27	0.95 - 1.7	.1	1.27	0.95 - 1.7	.1	1.45	1.14 - 1.85	.003	1.45	1.14 - 1.85	.003
J01FG01	Pristinamycin	1.90	1.39 - 2.61	<.001	1.90	1.38 - 2.6	<.001	1.65	1.25 - 2.17	<.001	1.64	1.25 - 2.16	<.001
N05BA04	Oxazepam	1.30	1.15 - 1.47	<.001	1.30	1.15 - 1.48	<.001	1.22	1.09 - 1.36	<.001	1.22	1.09 - 1.36	<.001
N05BA12	Alprazolam	1.07	0.98 - 1.18	.14	1.07	0.97 - 1.18	.16	1.12	1.04 - 1.2	.002	1.12	1.04 - 1.2	.002
N05BB01	Hydroxyzine	1.21	1.07 - 1.38	.003	1.21	1.07 - 1.38	.003	1.18	1.06 - 1.31	.002	1.18	1.06 - 1.31	.003
N06AX03	Mianserin	1.36	1.14 - 1.62	<.001	1.35	1.14 - 1.61	<.001	1.23	1.05 - 1.44	.01	1.23	1.05 - 1.44	.01
S01KA02	Hypromellose	0.77	0.62 - 0.94	.01	0.77	0.62 - 0.94	.01	0.77	0.65 - 0.91	.002	0.77	0.65 - 0.91	.002

Table R3: Results of sensitivity analyses. Estimated average treatment effect (ATE, Cox hazard ratio) for overall survival (OS) and disease-free survival (DFS), along with its 95% confidence interval, and p-value for the 16 medications significantly associated with OS, DFS, or both, after adjustment for multiple testing, obtained with two alternative timeframes for the use of diagnosis codes in hospital discharge to identify comorbid conditions. *Abbreviations: ATC: Anatomical Therapeutic Chemical; BC: Breast Cancer; HR: Hazard Ratio; CI: Confidence Interval (Supplementary Table 1 in the revised manuscript).*

- 3. In the Discussion section, the authors state that “Sex hormones have previously been associated with an increased risk of BC” and cite references #48 and #49 in support of that sentence. However, reference #48 is about the use of hormone therapy in cancer survivors, and reference #49 does not focus on breast cancer incidence. I think that other references would be more appropriate, such as the one from the Collaborative Group on Hormonal Factors in Breast Cancer published in 2019 [doi: 10.1016/S0140-6736(19)31709-X] regarding observational data, and one or two references related to the WHI randomized controlled trial.**

We fully agree with the reviewer and acknowledge that the previously cited references (#48 and #49) may not be directly focused on BC incidence. To provide more accurate and relevant support for our statement, we have updated the manuscript with the recommended reference from the Collaborative Group on Hormonal Factors in Breast Cancer¹ and two pertinent references related to the Women's Health Initiative (WHI) randomized controlled trial^{2,3} (doi:10.1001/jama.2020.9482 and doi:10.1001/jamaoncol.2015.0494).

- 4. In the discussion section, the authors state: “Our results on vaginal estrogens are also consistent with previous findings of their association with a lower risk of all-cause mortality when used after BC” and cite references #50, 51 and 52 to support the sentence. However, in ref #50, exposure to vaginal estrogens is assessed before breast cancer occurrence, not after; and in ref #52, the outcome is not all-cause mortality but breast cancer mortality.**

The reviewer is correct. We acknowledge the inconsistency in our original references and apologize for the lack of clarity. We have clarified the sentence and added appropriate references. For the sake of completeness, we have also added a reference to a newly published observational study on this topic⁴ (doi:10.1001/jamaoncol.2023.4508), and pointed out that concerns on the safety of vaginal estrogen therapy after BC for patients currently treated with aromatase inhibitors have been raised.

Page 12, line 314,

“Our results on vaginal estrogens are also consistent with previous findings of their association with a lower risk of all-cause mortality when used after BC⁵¹⁻⁵³.”

was replaced by

“Conversely, post-diagnosis vaginal estrogen therapy has been tentatively associated with a lower risk of BC recurrence⁵, specific mortality⁴, or all-cause mortality⁶, although safety concerns have been raised in patients currently treated with aromatase inhibitors⁶.”

- 5. When discussing their results about prednisolone, the authors state: “Our study revealed that prednisolone was associated with a 58% decrease in DFS and a 78%**

decrease in OS, in line with several preclinical studies suggesting that glucocorticoids may promote BC progression and metastasis (59–61)”. Besides preclinical studies, a recent epidemiological study is also in line with these results [DOI: 10.1186/s12916-021-02004-6]: in that cohort study, the use of glucocorticoids was associated with decreased risk of stage 1 or stage 2 tumours but increased risk of stage 3 or 4 cancers.

We appreciate the reviewer for bringing this reference to our attention and apologize for the oversight. We have now included a discussion of this reference in the manuscript.

Page 14, line 363,

“[...] and with an epidemiologic study reporting that the use of glucocorticoids was associated with a decreased risk of stage I-II BCs but an increased risk of stage III-IV BCs⁷”

was added.

⇒ ***We thank reviewer #2 for the time invested in reviewing our answers and for the remarks which enriched our analysis and made our manuscript gain nuance, precision and clarity.***

References

1. Collaborative Group on Hormonal Factors in Breast Cancer (2019). Type and timing of menopausal hormone therapy and breast cancer risk: individual participant meta-analysis of the worldwide epidemiological evidence. *Lancet Lond. Engl.* *394*, 1159–1168. [10.1016/S0140-6736\(19\)31709-X](https://doi.org/10.1016/S0140-6736(19)31709-X).
2. Chlebowski, R.T., Anderson, G.L., Aragaki, A.K., Manson, J.E., Stefanick, M.L., Pan, K., Barrington, W., Kuller, L.H., Simon, M.S., Lane, D., et al. (2020). Association of Menopausal Hormone Therapy With Breast Cancer Incidence and Mortality During Long-term Follow-up of the Women’s Health Initiative Randomized Clinical Trials. *JAMA* *324*, 369–380. [10.1001/jama.2020.9482](https://doi.org/10.1001/jama.2020.9482).
3. Chlebowski, R.T., Rohan, T.E., Manson, J.E., Aragaki, A.K., Kaunitz, A., Stefanick, M.L., Simon, M.S., Johnson, K.C., Wactawski-Wende, J., O’Sullivan, M.J., et al. (2015). Breast Cancer After Use of Estrogen Plus Progestin and Estrogen Alone: Analyses of Data From 2 Women’s Health Initiative Randomized Clinical Trials. *JAMA Oncol.* *1*, 296–305. [10.1001/jamaoncol.2015.0494](https://doi.org/10.1001/jamaoncol.2015.0494).
4. McVicker, L., Labeit, A.M., Coupland, C.A.C., Hicks, B., Hughes, C., McMenamin, Ú., McIntosh, S.A., Murchie, P., and Cardwell, C.R. (2023). Vaginal Estrogen Therapy Use and Survival in Females With Breast Cancer. *JAMA Oncol.*, e234508. [10.1001/jamaoncol.2023.4508](https://doi.org/10.1001/jamaoncol.2023.4508).
5. Durna, E.M., Leader, L.R., Sjoblom, P., Eden, J.A., Wren, B.G., and Heller, G.Z. (2002). Hormone replacement therapy after a diagnosis of breast cancer: cancer recurrence and mortality. *Med. J. Aust.* *177*, 347–351. [10.5694/j.1326-5377.2002.tb04835.x](https://doi.org/10.5694/j.1326-5377.2002.tb04835.x).
6. Cold, S., Cold, F., Jensen, M.-B., Cronin-Fenton, D., Christiansen, P., and Ejlersen, B. (2022). Systemic or Vaginal Hormone Therapy After Early Breast Cancer: A Danish Observational Cohort Study. *JNCI J. Natl. Cancer Inst.*, djac112. [10.1093/jnci/djac112](https://doi.org/10.1093/jnci/djac112).
7. Cairat, M., Al Rahmoun, M., Gunter, M.J., Heudel, P.-E., Severi, G., Dossus, L., and Fournier, A. (2021). Use of systemic glucocorticoids and risk of breast cancer in a prospective cohort of postmenopausal women. *BMC Med.* *19*, 186. [10.1186/s12916-021-02004-6](https://doi.org/10.1186/s12916-021-02004-6).

Reviewer #5:

As requested, I have reviewed the degree to which the authors have responded to the comments from referees #3 and #4.

We thank reviewer #5 for the time invested and the constructive comments on our previous responses. We have specifically addressed each concern below.

- 1. Reviewer 3's main concern was that the study was based upon medications used before diagnosis to infer the impact of medications after diagnosis on breast cancer-specific mortality. I think reviewer 3 is making an important point. In response the authors removed and added text to clarify that pre-diagnosis medications were being investigated e.g. "nor do they reflect the effect of medication use after cancer diagnosis or in a non-cancer population." They also changed the conclusion to: "This resource can guide the choice of candidates for drug repurposing trials or pharmacovigilance warnings." This still seems to apply to me that they are inferring the effect of the medication before diagnosis will be the same as after diagnosis. For instance, you cannot change the medications you have been using before your breast cancer diagnosis – so it is hard to see this as useful information for pharmacovigilance. And surely repurposing trials should be informed by medication use after breast cancer diagnosis.**

We thank the reviewer for the comment and apologize for the misunderstanding that our initial conclusions may have conveyed. We would like to clarify that our intention was not draw a direct analogy between the effect of pre-diagnosis medication and post-diagnosis medication, nor to suggest immediate implications for pharmacovigilance or drug repurposing based solely on pre-diagnosis drug exposure.

Investigating the influence of pre-diagnosis medication use on BC characteristics and prognosis offers valuable research hypotheses. Such an investigation could elucidate how chronic exposure to medications may influence the development and initial presentation of BC. Specifically, it may reveal variations in tumor subtypes, extent of lymph node involvement, and overall prognostic influenced by the "pressure" of long-term medication use. Understanding these dynamics may shed light on the ways in which pharmacological factors may contribute to the heterogeneity observed in breast cancer at the time of diagnosis.

We recognize the need for precision in our conclusion. We have revised it to better reflect the scope and implications of our findings, ensuring that our conclusions align with the hypothesis-generating nature of our findings, as follows.

In the discussion section, page 16 line 425

“This resource can guide the choice of candidates for drug repurposing trials or pharmacovigilance warnings.”

was replaced by

“This resource can also provide new hypotheses for drugs that may naturally BC evolution, from its presentation at diagnosis (subtype, lymph node involvement) to its long-term prognosis (overall survival, disease-free survival).”

In the abstract, page 3 line 67

“This resource can guide the choice of candidates for drug-repurposing trials or pharmacovigilance warnings.”

was replaced by

“This resource provides hypotheses for drugs that may naturally influence breast cancer evolution.”

- 2. Reviewer 3 was also concerned about the lack of analysis of dose response and the use of a binary 3 month use in 6 month definition. I think I would agree a dose response analysis is a non-trivial omission and dose response analyses are common in this kind of screening study (e.g., doi: 10.1016/j.ebiom.2016.03.018.). The authors have at least added this as a limitation.**

Thank you. We acknowledge the reviewer’s concern regarding the absence of dose-response analysis in our study. We concur that a dose-response analysis could refine our understanding of the identified associations and potentially uncover gradients of effect not captured by a binary categorization. While this analysis was beyond the scope of the current study due to the size of the dataset and the methodological challenges it presents, we recognize this as a limitation and as an important area for future research. We have further clarified this limitation in the revised manuscript as follows.

In the discussion section, page 15, line 407,

“Finally, while our study provides foundational insights into the association between pre-diagnosis medication use and BC outcomes, we recognize that the absence of dose-response analysis limits the granularity of our findings. Future investigations are needed to determine the potential dose-dependent effects of pre-diagnosis medications on disease progression and patient survival.”

was added.

- 3. Reviewer 3 was also concerned about the use of all-cause mortality rather than breast cancer-specific mortality. Here the authors have been responsive and have**

added disease free survival but not breast cancer specific mortality. They now mention this as a “main outcome”. I am not sure changing the main outcome is appropriate or transparent as their main outcome should be the main outcome of their original proposal.

We appreciate the reviewer’s concerns regarding the outcomes assessed in our study. The original design of our study included all-cause mortality as a primary endpoint. Due to the limited availability of cause of death data in our dataset, we could not examine breast cancer-specific mortality as an additional outcome, but we instead included disease-free survival. The inclusion of disease-free survival provides a more nuanced understanding of the association of concomitant medications with disease progression, which may be obscured when considering all-cause mortality alone.

We acknowledge the reviewer's point about the transparency and appropriateness of changing primary outcomes *post hoc*. To address this, we have now retained overall survival as the primary outcome while considering disease-free survival as a secondary outcome. Specifically, we have now systematically placed overall survival before disease-free survival throughout the manuscript. We have also clarified the objective of the study in the introduction, and the respective role of the two outcomes in the methods section.

Page 5, line 103,

“The main objective of this study, designated ADRENALINE (Atlas of Drugs, Comorbidities and Cancer Treatment Survival Interaction), is to analyze the impact of medication use in the six months preceding the diagnosis of BC on disease-free survival (DFS) and overall survival (OS) in a very large cohort of French women diagnosed with BC.”

was replaced by

*“The main objective of this study, designated ADRENALINE (Atlas of Drugs, Comorbidities and Cancer Treatment Survival Interaction), is to analyze the impact of medication use in the six months preceding the diagnosis of BC on **overall survival (OS, main outcome), and disease-free survival (DFS, secondary outcome)** and in a very large cohort of French women diagnosed with BC.”*

Page 19, line 503

“The main outcomes were disease-free survival (DFS) and overall survival (OS)”

was replaced by

“The primary endpoint was overall survival (OS). Disease-free survival (DFS) was evaluated as a secondary endpoint.”

- 4. Reviewer 3 discussed missing confounders including stage. The authors argue it is not a confounder but a mediator – which seems a reasonable argument. The authors**

though focus part of their analysis on mediation so missing stage remains an important omission. Surely if any of these medications impacted outcomes it could do so through stage and if mediation was an important goal of the study you would want to investigate mediation by stage.

Thank you for the comment. We concur that exploring how medications might impact BC outcomes through stage would be a valuable addition. Due to limitations in the data availability, we were unable to conduct a direct mediation analysis by stage.

However, the absence of stage in our analyses biases neither the ATE estimation, because stage is not a confounder but a mediator, nor the mediation analysis, because the effect of medication attributable solely to a change in tumor stage, if present, would simply be included in the direct path-specific effect (*i.e.*, the effect attributable neither to a change in BC subtype nor to a change in lymph node involvement).

In response to your comment, we further clarified the impact the absence of stage may have on the mediation analysis, as follows.

In the discussion section, page 16, line 416,

“Similarly, while we estimated the magnitude of the observed effect attributable to a change in BC subtype or lymph node involvement at diagnosis, we did not perform such mediation analysis for other BC biological characteristics such as tumor stage, nor for post-diagnosis drug use, whose mediation portion remained included in the direct path-specific effect.”

was replaced by

*“Similarly, while we estimated the magnitude of the observed effect attributable to a change in BC subtype or lymph node involvement at diagnosis, we did not perform such mediation analysis for other BC biological characteristics **such as tumor stage**, nor for post-diagnosis drug use, whose mediation portion remained included in the direct path-specific effect.”*

In general, they do not discuss other potential confounders in any real detail. For instance, BMI, smoking, alcohol intake, physical activity etc.

We appreciate the reviewer's feedback, and we acknowledge that the initial manuscript may not have sufficiently detailed the potential unmeasured confounders. Indeed, despite our comprehensive adjustment for approximately one hundred confounders, including comorbid conditions, socio-demographic characteristics, and prior medication use, we concur that our study might still be subject to residual confounding by factors not directly available in our dataset, such as body mass index (BMI), physical activity, smoking, alcohol intake, and other lifestyle habits or socioeconomic factors.

To address these variables indirectly, our analysis included proxies for these factors whenever possible, such as severe obesity as a proxy for BMI, severe tobacco and alcohol dependence for smoking and alcohol consumption, respectively, diabetes as an indicator of dietary habits, pulmonary disease for exposure to pollutants, and hypertension or cardiovascular diseases as indicators of stress levels. While these proxies provide insights into some of the most severe effects of these lifestyle factors, we recognize they may not capture the full spectrum of potential confounding. In addition, the addition of disease-free survival as a secondary outcome may have partially mitigated the influence of confounders that might affect overall survival through mechanisms not directly related to breast cancer relapse.

In light of the reviewer's comment, we have enriched the discussion section to more thoroughly address the potential impact of these unmeasured confounders on our findings, as follows.

Page 15, line 401,

“These include lifestyle and behavioral factors such as diet, body mass index, smoking, alcohol consumption, or physical activity. While the presence of proxy indicators in our adjustment set (e.g., severe obesity, diabetes, hypertension, severe tobacco and alcohol dependence, deprivation index) may mitigate this limitation, we acknowledge the importance of direct measures of these variables in future research.”

was added.

5. Reviewer 3 was concerned that the figures were hard to read. I would have to agree and I don't think the figures have changed much.

We apologize for the lack of clarity of our revised Figures. Reviewer #3 was specifically concerned about the readability of Figure 1 (distribution of comorbid conditions) and Figure 2 (distribution of concomitant medications).

We acknowledge that the labels of some comorbid conditions in Figure 1 were difficult to read. To improve the clarity of the figure, we regrouped all comorbid conditions affecting less than 2,000 units into "Other" for each category and listed the specific composition of these newly created "Other" groups in the figure legend. Of note, the Other group in the "Endocrine and metabolism" category (colored in orange) was still difficult to read. In response, we moved the label outside of the dedicated rectangle. We also increased the size of the legend labels. The updated figure is shown in Figure R1.

Regarding Figure 2, the purpose of the figure is to visually assess the overall distribution of concomitant medications and, in particular, to highlight the imbalance, with some

medications being used much more frequently than others (*e.g.*, colecalciferol, paracetamol). Readers interested in the specific number of users for each of the 288 molecules studied can refer to Supplementary Data or to the online platform, as specified in the Figure's legend. However, in response to your concerns about the figure's readability, we have removed the rings for ATC levels 3 and 4 from the sunburst. The updated figure is shown as Figure R2.

Figure R1: Distribution of comorbid conditions (by disease) in the total population. Diseases are color-coded by category. Percentages of the total population are reported. In each category, comorbid conditions with fewer than 2,000 cases were regrouped into the "Other" category to improve readability. In the neurologic and psychiatric diseases category, "Other" includes anorexia or bulimia (n=114, 0%), cognitive disabilities (n=791, 0.3%), epilepsy (n=1278, 0.5%), hemiplegia, paraplegia or palsy (n=1600, 0.7%), multiple sclerosis (n=719, 0.3%), other substance use disorder (n=244, 0.1%), and Parkinson's disease (n=989, 0.4%). In the cardiovascular diseases category, "Other" includes coagulopathy (n=743, 0.3%), hemoglobinopathy (n=104, 0%), and pulmonary embolism (n=1217, 0.5%). In the gastrointestinal diseases category, "Other" includes inflammatory bowel disease (n=1022, 0.4%), pancreatic disease (n=232, 0.1%), and peptic ulcer disease (n=576, 0.2%). In the endocrine and metabolic diseases category, "Other" includes other endocrine disorders (n=541, 0.2%). In the rheumatologic and connective tissue disorders category, "Other"

includes connective tissue diseases (n=1102, 0.5%), fibromyalgia (n=324 0.1%), osteoporosis (n=1817, 0.8%), and rheumatic diseases (n=664, 0.3%). In the other diseases category, “Other” includes hereditary metabolic disorders (n=459, 0.2%), myopathies, or disorders of muscles (n=562, 0.2%), HIV/AIDS (n=316, 0.1%), organ or tissue transplant (n=186 0.1%), other immune deficiency (n=141, 0.1%), chronic hepatitis (n=1001, 0.4%), cirrhosis (n=879, 0.4%), and steatosis and hereditary diseases (n=1046, 0.4%). The data can be further explored on the interactive ADRENALINE web application (https://adrenaline.curie.fr/comor_treemap, https://adrenaline.curie.fr/comorbidity_description). Abbreviations: HIV: human immunodeficiency virus; AIDS: acquired immunodeficiency syndrome (*Figure 1 in the revised manuscript*).

Figure R2: Distribution of concomitant medications by ATC code, for ATC level 1 (inner ring), ATC level 2 (middle ring), and ATC level 5 (outer ring). Concomitant medications are color-coded by ATC level. Raw data for ATC classes for which the data cannot be read on the graph can be accessed in Supplementary Data or via the interactive display available online at https://adrenaline.curie.fr/comed_description (Figure 2 in the revised manuscript).

- 6. I would agree with many of Reviewer 4's concerns. The cohort is poorly described – and this has not improved much e.g. I could not find a simple table of breast cancer characteristics anywhere – this is a strange omission. [...]**

We appreciate the reviewer's feedback on the need for a more detailed description of the BC cohort characteristics within the main body of the manuscript. In response, we have transferred the table that summarizes the patient sociodemographic characteristics, comorbid conditions, tumor biology, and BC treatments received from the supplementary materials to the main manuscript. This table, previously listed as Supplementary Table 1, has now been relabeled as Table 1. It is displayed here as Table R1.

Also I agree that the methods are (overly) complicated and not well described and this has not improved much. I think if you are going to publish this, an expert in causal analysis should thoroughly review their methods.

We thank the reviewer for the comment, and we apologize for the lack of clarity in the description of our methods. Our intention was to adopt methodologies offering straightforward causal interpretations, but we acknowledge that the initial presentation might not have been clear enough for all audience members.

To address these concerns, we have undertaken a comprehensive revision of the methods section in our manuscript. Specifically, we have detailed each step of the analysis in a more reader-friendly manner. Additionally, to offer a practical and visual guide to our causal inference process, we have moved the figure that outlines the five steps of the causal inference pipeline, from the supplementary material (formally Supplementary Figure 10) to the main body of the manuscript (now Figure 9). This figure, displayed here as Figure R3, is intended to visually complement the textual description and to provide a step-by-step example for one molecule (ranitidine). We have also retained detailed mathematical formulations, notations, and an extensive list of references in the supplementary material for readers interested in a deeper dive into the methodologies applied. We hope these adjustments will make the methods section more accessible and informative, thereby addressing the concerns raised.

In the methods, page 20, line 530

“It has five steps: (i) adjustment by inverse probability of treatment weighting on the pre-exposure covariates identified as confounding factors in the DAG analysis⁷⁰; (ii) adjustment quality check based on the standardized mean differences for each of the pre-exposure covariates after adjustment⁷⁰; (iii) ATE estimation by hazard ratio (HR) calculation for a univariate Cox proportional hazards model fitted to the weighted population for both outcomes⁷¹⁻⁷³, (iv) mediation analyses for BC subtype and nodal status at diagnosis for

medications with a significant ATE (Supplementary Fig. 11), and (v) weighted Kaplan-Meier survival curves for the medications with a significant ATE after the Benjamini-Hochberg (BH) multiple testing procedure⁷⁴. Further details on the methods used are provided in the Supplementary Notes.”

was replaced by

“It could be broken down into five steps. Further details on the methods used are provided in the Supplementary Notes.

Step 1: Adjustment by inverse probability of treatment weighting (IPTW)

We used inverse probability of treatment weighting (IPTW) to adjust for the confounding bias induced by pre-exposure covariates identified as confounding factors in the DAG analysis¹. This procedure involved: (i) estimating propensity scores (PS) i.e. the probability of receiving the drug concerned given the value of the pre-exposure covariates for each patient; (ii) weighting the dataset by assigning each patient a stabilized inverse probability weight derived from the PS. All subsequent analyses were conducted on the weighted dataset.

Step 2: Adjustment quality check

We checked the adjustment quality a posteriori by calculating the standardized mean differences (SMDs) for each pre-exposure covariate after adjustment¹. In accordance with published results², the adjustment quality was considered insufficient if any SMD had an absolute value above 0.1, in which case the molecule was discarded from subsequent analyses.

Step 3: Average treatment effect estimation

We estimated the average treatment effect (ATE) by calculating the hazard ratio (HR) of a univariate Cox proportional hazards model fitted to the weighted population³⁻⁵. We used Wald tests calculated with robust covariances to draw statistical inferences about the estimated HR. The threshold for statistical significance was $p = 0.05$.

Step 4: Mediation analyses

Molecules with a significant ATE were selected for mediation analyses, which involved breaking down the ATE into several pathways passing through two potential mediators, BC subtype and nodal status (Supplementary Fig. 11). We assumed BC subtype and nodal status to be causally related. Standard direct and indirect effects were not, therefore, directly identifiable for each mediator⁶. It was, nevertheless, possible to break the ATE down into three path-specific effects (PSEs): (1) the effect through pathways involving neither a difference in BC subtype nor in nodal status (direct effects); (2) the effect through pathways involving a difference in nodal status only (effect through node); (3) the effect through pathways involving a difference in BC subtype (and potentially involving a difference in nodal status; effect through subtype). PSEs were estimated by a weighting approach⁷ and are expressed as percentages of the total effect (which may be negative).

Step 5: Kaplan-Meier survival curves

Weighted Kaplan-Meier survival curves were plotted for the molecules with a significant ATE after the Benjamini-Hochberg (BH) multiple testing procedure⁸, and compared with an adjusted log-rank test⁹. The threshold for statistical significance was set at $p = 0.1$, due to the low power of adjusted log-rank tests⁵.”

Category	Variable	Class	Total	No concomitant medication	At least one drug	p
			235 368 (100%)	56 510 (24%)	178 858 (76%)	
Pre-exposure covariates						
Sociodemographic	Age at diagnosis (years)		60.0 [50.0, 69.0]	52.0 [45.0, 61.0]	63.0 [53.0, 71.0]	<0.001***
	Age at diagnosis (years, classes)	<30	1 124 (0.5)	477 (0.8)	647 (0.4)	<0.001
		30-39	10 539 (4.5)	4 631 (8.2)	5 908 (3.3)	
		40-49	43 206 (18.4)	17 582 (31.1)	25 624 (14.3)	
		50-59	58 003 (24.6)	18 297 (32.4)	39 706 (22.2)	
		60-69	64 042 (27.2)	10 994 (19.5)	53 048 (29.7)	
		70-79	39 163 (16.6)	3 458 (6.1)	35 705 (20.0)	
	Deprivation index (quintiles)	80+	19 291 (8.2)	1 071 (1.9)	18 220 (10.2)	
		1st quintile (least deprived)	46 323 (19.7)	12 369 (21.9)	33 954 (19.0)	<0.001
		2nd quintile	46 688 (19.8)	11 657 (20.6)	35 031 (19.6)	
3rd quintile		45 984 (19.5)	11 021 (19.5)	34 963 (19.5)		
4th quintile		46 183 (19.6)	10 504 (18.6)	35 679 (19.9)		
Overseas départements	45 992 (19.5)	9 538 (16.9)	36 454 (20.4)			
Overseas départements	4 198 (1.8)	1 421 (2.5)	2 777 (1.6)			
GP consultations*		5.0 [3.0, 9.0]	2.0 [1.0, 4.0]	6.0 [4.0, 10.0]	<0.001***	
GP consultations* (classes)	0	12 964 (5.5)	8 421 (14.9)	4 543 (2.5)	<0.001	
	1-5	109 539 (46.5)	38 489 (68.1)	71 050 (39.7)		
	6-11	32 701 (13.9)	1 308 (2.3)	31 393 (17.6)		
	12+	80 164 (34.1)	8 292 (14.7)	71 872 (40.2)		
Gynecologist visits**		0.0 [0.0, 1.0]	1.0 [0.0, 1.0]	0.0 [0.0, 1.0]	<0.001***	
Gynecologist visits** (classes)	0	127 785 (54.3)	28 225 (49.9)	99 560 (55.7)	<0.001	
	1	65 868 (28.0)	17 694 (31.3)	48 174 (26.9)		
	2-3	35 290 (15.0)	8 941 (15.8)	26 349 (14.7)		
	4+	6 425 (2.7)	1 650 (2.9)	4 775 (2.7)		
Mammographic screening before diagnosis	No	146 945 (62.4)	38 843 (68.7)	108 102 (60.4)	<0.001	
Yes	88 423 (37.6)	17 667 (31.3)	70 756 (39.6)			
Comorbid conditions	Comorbid conditions (binary)	No	124 652 (53.0)	44 872 (79.4)	79 780 (44.6)	<0.001
		Yes	110 716 (47.0)	11 638 (20.6)	99 078 (55.4)	
	Comorbid condition category	Cardiovascular	60 146 (25.6)	2 931 (5.2)	57 215 (32.0)	<0.001
		Endocrine and metabolism	51 588 (21.9)	3 522 (6.2)	48 066 (26.9)	<0.001
		Psychiatric disorders	30 372 (12.9)	4 713 (8.3)	25 659 (14.3)	<0.001
		Frailty (proxy)	11 888 (5.1)	1 181 (2.1)	10 707 (6.0)	<0.001
		Pulmonary	10 883 (4.6)	750 (1.3)	10 133 (5.7)	<0.001
		Rheumatologic disease and connective tissue diseases	7 918 (3.4)	413 (0.7)	7 505 (4.2)	<0.001
		Gastrointestinal	7 519 (3.2)	752 (1.3)	6 767 (3.8)	<0.001
		Neurologic	6 983 (3.0)	746 (1.3)	6 237 (3.5)	<0.001
		Liver	2 668 (1.1)	324 (0.6)	2 344 (1.3)	<0.001
		Kidney	2 524 (1.1)	93 (0.2)	2 431 (1.4)	<0.001
		Other	1 015 (0.4)	103 (0.2)	912 (0.5)	<0.001
		Immune	635 (0.3)	84 (0.1)	551 (0.3)	<0.001
		Post-exposure covariates				
BC biology	Inferred BC subtype	luminal	153 109 (65.1)	34 117 (60.4)	118 992 (66.5)	<0.001
		TNBC	18 149 (7.7)	5 532 (9.8)	12 617 (7.1)	
		HER2+	19 722 (8.4)	5 974 (10.6)	13 748 (7.7)	
		Undefined	44 388 (18.9)	10 887 (19.3)	33 501 (18.7)	
Nodal status	Node-negative	191 164 (81.2)	45 282 (80.1)	145 882 (81.6)	<0.001	
	Node-positive	44 204 (18.8)	11 228 (19.9)	32 976 (18.4)		
BC treatment	Breast surgery	Partial mastectomy	173 173 (73.6)	40 238 (71.2)	132 935 (74.3)	<0.001
		Mastectomy	62 195 (26.4)	16 272 (28.8)	45 923 (25.7)	
	Radiotherapy	No	34 683 (14.7)	8 294 (14.7)	26 389 (14.8)	0.657
		Yes	200 685 (85.3)	48 216 (85.3)	152 469 (85.2)	
	Chemotherapy	No	145 116 (61.7)	29 421 (52.1)	115 695 (64.7)	<0.001
		Yes	90 252 (38.3)	27 089 (47.9)	63 163 (35.3)	
Endocrine therapy	No	69 713 (29.6)	18 628 (33.0)	51 085 (28.6)	<0.001	
	Yes	165 655 (70.4)	37 882 (67.0)	127 773 (71.4)		

Table R1: Characteristics of the patients in the total population, patients without medication at the time of BC diagnosis, and patients on at least one medication at the time of BC diagnosis. The number of patients, and the percentage of patients (in parentheses), are reported for categorical variables. The median value, and the interquartile range (in parentheses), are reported for continuous variables. Wilcoxon-Mann-Whitney tests were used to compare continuous variables with non-normal distributions (denoted by ***). Student's t-tests were used to compare other continuous variables. Fisher's exact test was used to assess associations between categorical variables if at least one category included fewer than three patients. Chi-squared tests were used to assess associations between other categorical variables. *Number of general practitioner (GP) visits in the year preceding BC diagnosis. **Number of gynecologist visits in the year preceding BC diagnosis. *Abbreviations: GP: general practitioner; BC: breast cancer; TNBC: triple-negative breast cancer.* (Table 1 in the revised manuscript)

Figure R3: Causal inference pipeline of the study (left) and illustration for one medication (ranitidine, ATC code A02BA02, right). Step 1: All potential confounders were adjusted by inverse probability of treatment weighting (IPTW), which requires the calculation of propensity scores (PS). The distribution of PSs is displayed for ranitidine, for patients exposed

(green) and not exposed (gray) to this medication. Step 2: We checked adjustment quality by: (a) calculating the standardized mean differences (SMDs) of all potential confounders, (b) checking that all SMDs had an absolute value below 0.1. If at least one SMD had an absolute value of more than 0.1, the quality of the adjustment was considered insufficient, and the medication was discarded from further analyses. We display the SMDs for age and number of medications in comedication for ranitidine before adjustment (in blue) and after adjustment (in red). Ranitidine passed the adjustment quality test for these two variables. Step 3: Average treatment effect was estimated by deriving the hazard ratio (HR) from a weighted Cox proportional hazard model for both outcomes (OS and DFS) fitted to the weighted dataset and comparing it to 1 in a Wald test with a robust covariance. The Cox HR for ranitidine was 0.62 for OS (95% CI 0.41 to 0.92) and 0.79 for DFS (95% CI 0.54 to 1.14). Step 4: For the medications significantly associated with OS or DFS (HR $P < .05$), mediation analyses were performed to break down the ATE into three path-specific effects. For ranitidine, none of the PSEs was significant. Step 5: For the medications significantly associated with survival after multiple testing correction with the Benjamini-Hochberg procedure (HR $P_{corrected} < .05$), we plotted adjusted Kaplan-Meier survival curves. We display the adjusted survival curves for patients exposed to ranitidine (in green) versus patients not exposed to ranitidine (in gray). *Abbreviations: PS: propensity score; IPTW: inverse probability of treatment weighting; SMD: standardized mean difference; ATE: average treatment effect; HR: hazard ratio; ATC: Anatomical Therapeutic Chemical; CT: chemotherapy; TNBC: triple-negative breast cancer; OS: overall survival; DFS: disease-free survival* (Figure 9 in the revised manuscript).

- 7. Unrelated to the comments of the reviewers, a further weakness, not discussed, is that the cohort has quite short follow-up for mortality (of <5 years). Breast cancer can recur many years after diagnosis and I think throughout the authors should really refer to early death/recurrence.**

Thank you for the suggestion. We acknowledge that BC is a long-term evolving disease, that can recur many years after diagnosis¹⁰, necessitating long-term follow-up to fully understand survival outcomes.

However, the median time from initial BC surgery to recurrence was previously reported to be approximately 48 months (4 years)¹¹, which falls within our median follow-up of 53.9 months (approximately 4.5 years) for disease-free survival. Furthermore, BC recurrence rates are known to be most significant within the first five years after treatment¹², and patients with recurrence occurring after five years generally have a more favorable prognosis than those with early recurrence¹³. Overall, this provides a rationale for the importance of early survival patterns.

In response to your suggestion, we have added a sentence to the discussion section to clarify that our findings predominantly relate to early deaths and recurrences, as follows.

Page 15, line 405,

"[...] due to our limited follow-up period, with a median of four and a half years, our results relate predominantly to early deaths and recurrences."

was added.

- 8. The authors suggest estrogen (vaginal/transmucosal) is protective in breast cancer patients but they should really address the randomised controlled trials of estrogen (<https://doi.org/10.1093/jnci/djs014>) and tibolone (Lancet Oncol. 2009 Feb;10(2):135-46) which have shown increases in breast cancer recurrence in breast cancer patients using either medication.**

We appreciate the reviewer's feedback and the opportunity to clarify our findings in the context of the existing literature on estrogen use in BC patients. We recognize the importance of distinguishing our observations from those reported in randomized controlled trials, such as the WHI¹⁴ or the tibolone trial¹⁵. The WHI evaluated the effect of estrogen alone in postmenopausal women with prior hysterectomy or estrogen plus progestin in postmenopausal women with an intact uterus in the general population on several outcomes, including BC risk and BC-related death. The study concluded that estrogen plus progestin, but not estrogen alone, was associated with an increased risk of BC and BC-related death in the overall population, including women who will not develop BC¹⁶. The tibolone study examined

the effect of tibolone use after diagnosis in BC patients and documented increased BC recurrence in tibolone users after diagnosis.

Our results suggest that use of sex hormones prior to BC diagnosis was associated with longer survival in BC patients, and that this protective association was at least in part due to a decreased likelihood of lymph node involvement at BC diagnosis with vaginal/transmucosal estrogen. Thus, our results are not inconsistent with those of the WHI randomized trial of estrogen, which evaluated BC-related deaths in all units regardless of whether they develop BC, or with those of the tibolone randomized trial, which evaluated the effect of initiating tibolone after BC diagnosis.

Overall, the literature on sex hormones and BC risk, biology, and survival presents inconsistent results, with distinctions based on hormone type (estrogen alone vs. combined with progestin)¹⁶, origin (synthetic vs. natural)¹⁷, route of administration (systemic vs. local)¹⁸, and subsequent endocrine treatments¹⁸. Thus, the effect of estrogen/progestin on BC risk, biology, and survival is complex and not fully understood to date, and our results may contribute to the body of evidence on this topic.

In light of your feedback, we have revised our discussion to better articulate these nuances, as follows:

Page 12, line 308,

“Several sex hormones used either locally (vaginal or transmucosal estriol treatment) or systemically (nomegestrol) were associated with longer survival, with HRs of 0.58 and 0.39 for OS, and 0.80 and 0.74 for DFS, respectively. Sex hormones have previously been associated with an increased risk of BC^{19–21}. Our results suggest that the protective effect of sex hormones on mortality in a population of patients with BC may be partly due to changes in tumor biology at diagnosis, with 23.8% of our estimated protective association of vaginal or transmucosal estriol with DFS mediated by an indirect effect through a decreased likelihood of lymph node involvement at BC diagnosis.”

was replaced by

“Several sex hormones used either locally (vaginal or transmucosal estriol treatment) or systemically (nomegestrol) were associated with longer survival in our cohort of BC patients when used prior to diagnosis, with HRs of 0.58 and 0.39 for OS, and 0.80 and 0.74 for DFS, respectively. The relationship between sex hormones and BC progression remains incompletely understood. Sex hormones used systematically have previously been associated with an increased risk of BC^{48–50}, or with an increased risk of relapse when use post-diagnosis⁵¹. Conversely, post-diagnosis vaginal estrogen therapy has been tentatively associated with a lower risk of BC recurrence⁵², specific mortality⁵³, or all-cause mortality⁵⁴, although safety concerns have been raised in patients currently treated with aromatase inhibitors⁵⁴. While our

findings do not address the risk of BC incidence or the impact of post-diagnosis hormone use, they suggest that pre-diagnosis sex hormones may be associated with decreased mortality and relapse in BC patients, possibly through changes in tumor biology at diagnosis, as 23.8% of the protective association we observed between vaginal or transmucosal estriol and DFS was mediated by a decreased likelihood of lymph node involvement.”

References

1. Austin, P.C., and Stuart, E.A. (2015). Moving towards best practice when using inverse probability of treatment weighting (IPTW) using the propensity score to estimate causal treatment effects in observational studies. *Stat. Med.* 34, 3661–3679. 10.1002/sim.6607.
2. Austin, P.C. (2011). An Introduction to Propensity Score Methods for Reducing the Effects of Confounding in Observational Studies. *Multivar. Behav. Res.* 46, 399–424. 10.1080/00273171.2011.568786.
3. Cole, S.R., and Hernán, M.A. (2004). Adjusted survival curves with inverse probability weights. *Comput. Methods Programs Biomed.* 75, 45–49. 10.1016/j.cmpb.2003.10.004.
4. Austin, P.C., and Schuster, T. (2016). The performance of different propensity score methods for estimating absolute effects of treatments on survival outcomes: A simulation study. *Stat. Methods Med. Res.* 25, 2214–2237. 10.1177/0962280213519716.
5. Le Borgne, F., Giraudeau, B., Querard, A.H., Giral, M., and Foucher, Y. (2016). Comparisons of the performance of different statistical tests for time-to-event analysis with confounding factors: practical illustrations in kidney transplantation. *Stat. Med.* 35, 1103–1116. 10.1002/sim.6777.
6. Vanderweele, T.J. (2015). *Explanation in causal inference: Methods for mediation and interaction* (Oxford University Press).
7. VanderWeele, T.J., Vansteelandt, S., and Robins, J.M. (2014). Effect decomposition in the presence of an exposure-induced mediator-outcome confounder. *Epidemiol. Camb. Mass* 25, 300–306. 10.1097/EDE.0000000000000034.
8. Benjamini, Y., and Hochberg, Y. (1995). Controlling the False Discovery Rate: A Practical and Powerful Approach to Multiple Testing. *J. R. Stat. Soc. Ser. B Methodol.* 57, 289–300.
9. Chatton, A., Le Borgne, F., Leyrat, C., Gillaizeau, F., Rousseau, C., Barbin, L., Laplaud, D., Léger, M., Giraudeau, B., and Foucher, Y. (2020). G-computation, propensity score-based methods, and targeted maximum likelihood estimator for causal inference with different covariates sets: a comparative simulation study. *Sci. Rep.* 10, 9219. 10.1038/s41598-020-65917-x.
10. Early Breast Cancer Trialists' Collaborative Group (EBCTCG), Davies, C., Godwin, J., Gray, R., Clarke, M., Cutter, D., Darby, S., McGale, P., Pan, H.C., Taylor, C., et al. (2011). Relevance of breast cancer hormone receptors and other factors to the efficacy of adjuvant tamoxifen: patient-level meta-analysis of randomised trials. *Lancet Lond. Engl.* 378, 771–784. 10.1016/S0140-6736(11)60993-8.

11. Horan, J., Reid, C., Boland, M.R., Daly, G.R., Keelan, S., Lloyd, A.J., Downey, E., Walmsley, A., Staunton, M., Power, C., et al. (2023). Assessing Mode of Recurrence in Breast Cancer to Identify an Optimised Follow-Up Pathway: 10-Year Institutional Review. *Ann. Surg. Oncol.* *30*, 6117–6124. 10.1245/s10434-023-13885-7.
12. Wangchinda, P., and Ithimakin, S. (2016). Factors that predict recurrence later than 5 years after initial treatment in operable breast cancer. *World J. Surg. Oncol.* *14*, 223. 10.1186/s12957-016-0988-0.
13. Pedersen, R.N., Mellekjær, L., Ejlersen, B., Nørgaard, M., and Cronin-Fenton, D.P. (2022). Mortality After Late Breast Cancer Recurrence in Denmark. *J. Clin. Oncol.* *40*, 1450–1463. 10.1200/JCO.21.02062.
14. Design of the Women’s Health Initiative clinical trial and observational study. The Women’s Health Initiative Study Group (1998). *Control. Clin. Trials* *19*, 61–109. 10.1016/s0197-2456(97)00078-0.
15. Kenemans, P., Bundred, N.J., Foidart, J.-M., Kubista, E., von Schoultz, B., Sismondi, P., Vassilopoulou-Sellin, R., Yip, C.H., Egberts, J., Mol-Arts, M., et al. (2009). Safety and efficacy of tibolone in breast-cancer patients with vasomotor symptoms: a double-blind, randomised, non-inferiority trial. *Lancet Oncol.* *10*, 135–146. 10.1016/S1470-2045(08)70341-3.
16. Chlebowski, R.T., and Anderson, G.L. (2012). Changing Concepts: Menopausal Hormone Therapy and Breast Cancer. *JNCI J. Natl. Cancer Inst.* *104*, 517–527. 10.1093/jnci/djs014.
17. Asi, N., Mohammed, K., Haydour, Q., Gionfriddo, M.R., Vargas, O.L.M., Prokop, L.J., Faubion, S.S., and Murad, M.H. (2016). Progesterone vs. synthetic progestins and the risk of breast cancer: a systematic review and meta-analysis. *Syst. Rev.* *5*, 121. 10.1186/s13643-016-0294-5.
18. Cold, S., Cold, F., Jensen, M.-B., Cronin-Fenton, D., Christiansen, P., and Ejlersen, B. (2022). Systemic or Vaginal Hormone Therapy After Early Breast Cancer: A Danish Observational Cohort Study. *JNCI J. Natl. Cancer Inst.*, djac112. 10.1093/jnci/djac112.
19. Collaborative Group on Hormonal Factors in Breast Cancer (2019). Type and timing of menopausal hormone therapy and breast cancer risk: individual participant meta-analysis of the worldwide epidemiological evidence. *Lancet Lond. Engl.* *394*, 1159–1168. 10.1016/S0140-6736(19)31709-X.
20. Chlebowski, R.T., Anderson, G.L., Aragaki, A.K., Manson, J.E., Stefanick, M.L., Pan, K., Barrington, W., Kuller, L.H., Simon, M.S., Lane, D., et al. (2020). Association of Menopausal Hormone Therapy With Breast Cancer Incidence and Mortality During Long-term Follow-up of the Women’s Health Initiative Randomized Clinical Trials. *JAMA* *324*, 369–380. 10.1001/jama.2020.9482.

21. Chlebowski, R.T., Rohan, T.E., Manson, J.E., Aragaki, A.K., Kaunitz, A., Stefanick, M.L., Simon, M.S., Johnson, K.C., Wactawski-Wende, J., O'Sullivan, M.J., et al. (2015). Breast Cancer After Use of Estrogen Plus Progestin and Estrogen Alone: Analyses of Data From 2 Women's Health Initiative Randomized Clinical Trials. *JAMA Oncol.* *1*, 296–305. [10.1001/jamaoncol.2015.0494](https://doi.org/10.1001/jamaoncol.2015.0494).

REVIEWERS' COMMENTS

Reviewer #2 (Remarks to the Author):

The authors adequately addressed my comments.

Reviewer #3 (Remarks to the Author):

The authors have addressed to some extent most of my comments. I am not totally convinced that they have investigated medications at the most informative time point and I have still have some concerns about the short duration of follow-up. However, these are discussed.